*Report*

# A genome editing approach to study cancer stem cells in human tumors

Carme Cortina[1,†], Gemma Turon[1,†], Diana Stork[1], Xavier Hernando-Momblona[1], Marta Sevillano[1], Mònica Aguilera[1], Sébastien Tosi[1], Anna Merlos-Suárez[1], Camille Stephan-Otto Attolini[1], Elena Sancho[1] & Eduard Batlle[1,2,*]

## Abstract

The analysis of stem cell hierarchies in human cancers has been hampered by the impossibility of identifying or tracking tumor cell populations in an intact environment. To overcome this limitation, we devised a strategy based on editing the genomes of patient-derived tumor organoids using CRISPR/Cas9 technology to integrate reporter cassettes at desired marker genes. As proof of concept, we engineered human colorectal cancer (CRC) organoids that carry EGFP and lineage-tracing cassettes knocked in the LGR5 locus. Analysis of LGR5-EGFP[+] cells isolated from organoid-derived xenografts demonstrated that these cells express a gene program similar to that of normal intestinal stem cells and that they propagate the disease to recipient mice very efficiently. Lineage-tracing experiments showed that LGR5[+] CRC cells self-renew and generate progeny over long time periods that undergo differentiation toward mucosecreting- and absorptive-like phenotypes. These genetic experiments confirm that human CRCs adopt a hierarchical organization reminiscent of that of the normal colonic epithelium. The strategy described herein may have broad applications to study cell heterogeneity in human tumors.

**Keywords** cancer stem cells; colorectal cancer; CRISPR/Cas9; LGR5
**Subject Categories** Cancer; Digestive System; Stem Cells

See also: **SM Dieter *et al*** (July 2017)

## Introduction

Most cancers are amalgams of phenotypically distinct tumor cell populations, which display marked differences in their behaviors and fates. In colorectal cancer (CRC), a subpopulation of cells with elevated tumorigenic potential expresses a gene program similar to that of intestinal stem cells (ISCs). These ISC-like tumor cells give rise to differentiated-like progeny, which is poorly tumorigenic (Dalerba *et al*, 2007, 2011; O'Brien *et al*, 2007; Ricci-Vitiani *et al*, 2007; Vermeulen *et al*, 2008, 2010; Merlos-Suarez *et al*, 2011). These findings have led to the notion that CRCs retain a hierarchical organization reminiscent of that of the normal intestinal mucosa, with only cancer stem cells being capable of self-renewal and of sustaining long-term tumor growth (Zeuner *et al*, 2014). To a large extent, this model has emerged from experiments of tumor cell transplantation. Typically, putative stem and non-stem cell populations are isolated from patient samples using combinations of surface markers, and then, each cell population is inoculated into immunodeficient mice. The capacity to generate xenografts and to reproduce some of the traits of the tumor of origin are used as readouts of stemness. These assays, however, only provide a snapshot of the state of the cells in the moment they were isolated. It is also unclear to what extent experimental manipulations influence the tumor-initiating capacity of purified cells (Clevers, 2011). Furthermore, the requirement of antibodies against surface markers to isolate tumor cells from patient samples imposes limitations to explore the diversity of cell phenotypes within cancers. Alternatively, the existence of tumor stem cells has been confirmed in mouse adenomas through genetic fate-mapping experiments (Schepers *et al*, 2012; Kozar *et al*, 2013). Yet, these lesions are benign and contain few mutations compared to human CRCs. To overcome these restrictions, we combined CRC patient-derived organoids (PDOs) with CRISPR/Cas9 technology to label defined tumor cell populations and perform fate-mapping experiments *in vitro* and *in vivo*.

## Results

### Generation of LGR5-EGFP knock-in human CRC organoids

The expression of the Leucine-rich repeat-containing G-protein-coupled receptor 5 (LGR5) marks adult ISCs in mice and humans (Barker *et al*, 2007; Jung *et al*, 2011). Knock-in mice engineered to carry EGFP and CreERT2 cassettes integrated into the LGR5 locus have been instrumental to visualize and track ISCs in the healthy

1  Institute for Research in Biomedicine (IRB Barcelona), The Barcelona Institute of Science and Technology, Barcelona, Spain
2  Institució Catalana de Recerca i Estudis Avançats (ICREA) and CIBER-ONC, Barcelona, Spain
   *Corresponding author. Tel: +34 934039008; E-mail: eduard.batlle@irbbarcelona.org
   †These authors contributed equally to this work

mucosa and in tumors (Barker *et al*, 2007, 2009; Schepers *et al*, 2012). In contrast, the analysis of LGR5$^+$ cell populations in human cancers has been hampered by the lack of good commercial antibodies that recognize this protein at the cell surface. We thus designed a strategy based on CRISPR/Cas9-mediated homologous recombination to mark LGR5$^+$ cells in human CRCs. We made use of CRC PDOs, which are good surrogates of the disease *in vitro* and *in vivo* (Calon *et al*, 2015; van de Wetering *et al*, 2015). For these experiments, we initially selected a PDO derived from a stage IV CRC that displayed a prototypical combination of genetic alterations in major driver pathways including activation of the WNT pathway by APC loss of function, activation of EGFR signaling by KRAS G13D mutations, and loss of TGF-beta-mediated tumor suppression by inactivating mutations in SMAD4 (PDO#7 in Appendix Table S1). The targeting strategy is summarized in Fig 1A and detailed in the Materials and Methods section. In brief, we designed Cas9 guide RNAs complementary to sequences overlapping the stop codon of the LGR5 locus and generated a donor vector that contained LGR5 homology arms flanking an EGFP reporter cassette positioned immediately upstream of the stop codon. We added a LF2A self-cleavage peptide (de Felipe *et al*, 2010) fused to EGFP so that LGR5-EGFP locus was expressed as a single mRNA, whereas the resulting polypeptide was cleaved in the two encoded proteins, LGR5 and EGFP (Fig 1A). Next, we nucleofected organoid cells with the donor vector together with a guide-RNA-Cas9 encoding plasmid in a 3:1 proportion, and 48 h after, we sorted cells that had incorporated the Cas9 vector (IRFP$^+$ cells). About 1 in 11 IRFP$^+$ cells expressed EGFP after 20 days in culture (Fig 1B). Subsequently, we generated single cell-derived organoid cultures and assessed integration of the EGFP reporter cassette by PCR (examples in Appendix Fig S1A and B) and Southern blot (examples in Appendix Fig S1C and D). These analyses showed that 41.7% of the clones had correctly integrated the EGFP reporter in the LGR5 locus (Appendix Table S2). Equivalent LGR5-EGFP knock-in experiments in a PDO grown from a different patient sample (PDO#6) (Fig EV1A) rendered a frequency of correct integrations of 84.6% (Appendix Table S2). In these single cell-derived knock-in PDO cultures, every organoid was composed by an admixture of cells expressing distinct EGFP levels (Figs 1C and D, and EV1B and C). LGR5-EGFP-hi cells isolated by FACS expressed highest LGR5 mRNA levels confirming that EGFP reported endogenous LGR5 expression (Figs 1E and EV1D). Staining with KRT20 or MUC2 antibodies revealed complementary expression patterns of these differentiation markers with EGFP implying that LGR5$^+$ CRC cells generated differentiated progeny *in vitro* (Fig 1C).

To demonstrate the broad applicability of this approach, we also engineered PDO#7 expressing TagRFP2 fused to endogenous KI67 protein (Fig 1F). The KI67 antigen is a nuclear protein which is expressed in all active phases of the cell cycle (G1, S, G2, and mitosis), but is absent in resting cells (G0) (Scholzen & Gerdes, 2000). In a previous study, knock-in mice expressing a KI67-RFP fusion protein were used to isolate cycling (KI67-RFP$^+$) and non-cycling differentiated cells (KI67-RFP$^-$) from the intestinal epithelium (Basak *et al*, 2014). Targeting efficiency for this knock-in construct in human CRC organoids was similar to that observed for LGR5-EGFP knock-in organoids (Appendix Table S2). KI67-TagRFP2 was visualized in the nucleus of organoid cells (Fig 1G) and of xenografts derived from these organoids (Fig 1H). Cell cycle profiling of epithelial cells isolated using FACS from dissociated xenografts (Fig 1I) demonstrated that TagRFP2$^+$ cells were distributed in all cell cycle phases, whereas the TagRFP2$^-$ population was largely enriched in cells at the G1/G0 phase (Fig 1J and K).

## Characterization of human LGR5$^+$ CRC cells *in vivo*

To study LGR5$^+$ cells *in vivo*, we initially used an LGR5-EGFP expressing organoid (clone #1) that by exome sequencing revealed few acquired mutations compared to the parental population, none of which affected known cancer driver genes (Appendix Table S3). Clone #1 neither contained mutations in the top off-target sequences predicted through bioinformatics for the CRISPR guide sequence (Appendix Table S4). The LGR5-EGFP knock-in PDO was inoculated into immunodeficient mice of the NOD/SCID strain. Xenografts displayed a glandular organization and prominent stromal recruitment. LGR5-EGFP expression labeled a substantial proportion of the epithelial component of the tumor yet cells showed a wide range of EGFP levels (Figs 2A and EV2). In contrast, the EGFP$^-$ compartment overlapped largely with the expression domain of the pan differentiation marker KRT20$^+$ (Fig 2B). We also observed LGR5$^-$/MUC2$^+$ cells with goblet-like morphology intermingled between LGR5$^+$ and LGR5$^-$ compartments throughout the tumor (Fig 2C). These tumor cell populations displayed equivalent distributions in xenografts produced by a different single cell-derived clone from PDO#7 (Fig EV3A–C) or by PDO#6 (Fig EV1E–G). Overall, this cellular organization is reminiscent of that of the normal intestinal epithelium as previously proposed by several laboratories (Dalerba *et al*, 2011; Merlos-Suarez *et al*, 2011).

In flow cytometry analysis, LGR5-EGFP-high cells represented about 3–4% of the epithelial component (EPCAM$^+$) of dissociated

---

**Figure 1.   LGR5-EGFP and KI67-TagRFP2 knock-in PDOs.**

A   Design of LGR5-EGFP donor and CRISPR/Cas9 sgRNA vectors. Blue circle represents the CRISPR/Cas9 protein complex and the yellow box underneath the guide RNA.

B   Flow cytometry profiles at day 20 post-nucleofection.

C   Immunofluorescence for DAPI, EGFP, and KRT20 or MUC2 in *in vitro* cultured PDO#7-LGR5-EGFP#1. Scale bars indicate 100 μm.

D   FACS profiles showing EGFP-high (green), -low (blue), and -negative (gray) cells in PDO#7-LGR5-EGFP#1 and #2 organoids.

E   Relative mRNA expression level by real-time qPCR in cells expressing distinct levels of EGFP isolated from PDO#7-LGR5-EGFP#1 and #2 knock-in organoids. Values show mean ± s.d. of three measurements.

F   Design of KI67-TagRFP2 donor and CRISPR/Cas9 sgRNA vectors. Blue circle represents the CRISPR/Cas9 protein complex and the yellow box underneath the guide RNA.

G   Images of PDO#7-KI67-TagRFP2#1 organoids. Scale bars indicate 100 μm.

H   PDO#7-KI67-TagRFP2#1 xenograft. TagRFP2 co-localizes with DAPI nuclear staining. Scale bars indicate 25 μm.

I   Flow cytometry analysis of EPCAM$^+$/DAPI$^-$ cell population of PDO#7-LGR5-EGFP/KI67-TagRFP2#1 from disaggregated xenografts.

J   Cell cycle analysis of KI67-TagRFP2-positive and KI67-TagRFP2-negative cells from PDO#7-LGR5-EGFP/KI67-TagRFP2#1 disaggregated xenografts. *X*-axis shows DNA content and *y*-axis EdU incorporation.

K   Quantification of the frequencies of KI67$^+$ versus KI67$^-$ cells found in each cell cycle phase.

    

xenografts (Fig 2D). We isolated EGFP-high and EGFP-low/negative cells by FACS (for simplicity we termed them EGFP$^+$ and EGFP$^-$) and analyzed their global gene expression profiles. LGR5-EGFP$^+$ cells expressed over 10-fold higher levels of ISC marker genes LGR5 and SMOC2 than EGFP$^-$ cells (Fig 2E). LGR5-EGFP$^-$ cells expressed genes that characterize differentiated cells of the intestinal epithelium such

as EFNB2, KRT20 or MUC2 (Fig 2E). We validated these results using a second LGR5-EGFP knock-in clone derived from PDO#7 (Fig EV3D and E). Microarray profiling followed by gene set enrichment analysis (GSEA) confirmed that mouse and human intestinal stem cell gene expression signatures were upregulated in LGR5-EGFP$^+$, whereas the differentiation program of colon epithelium was enriched in

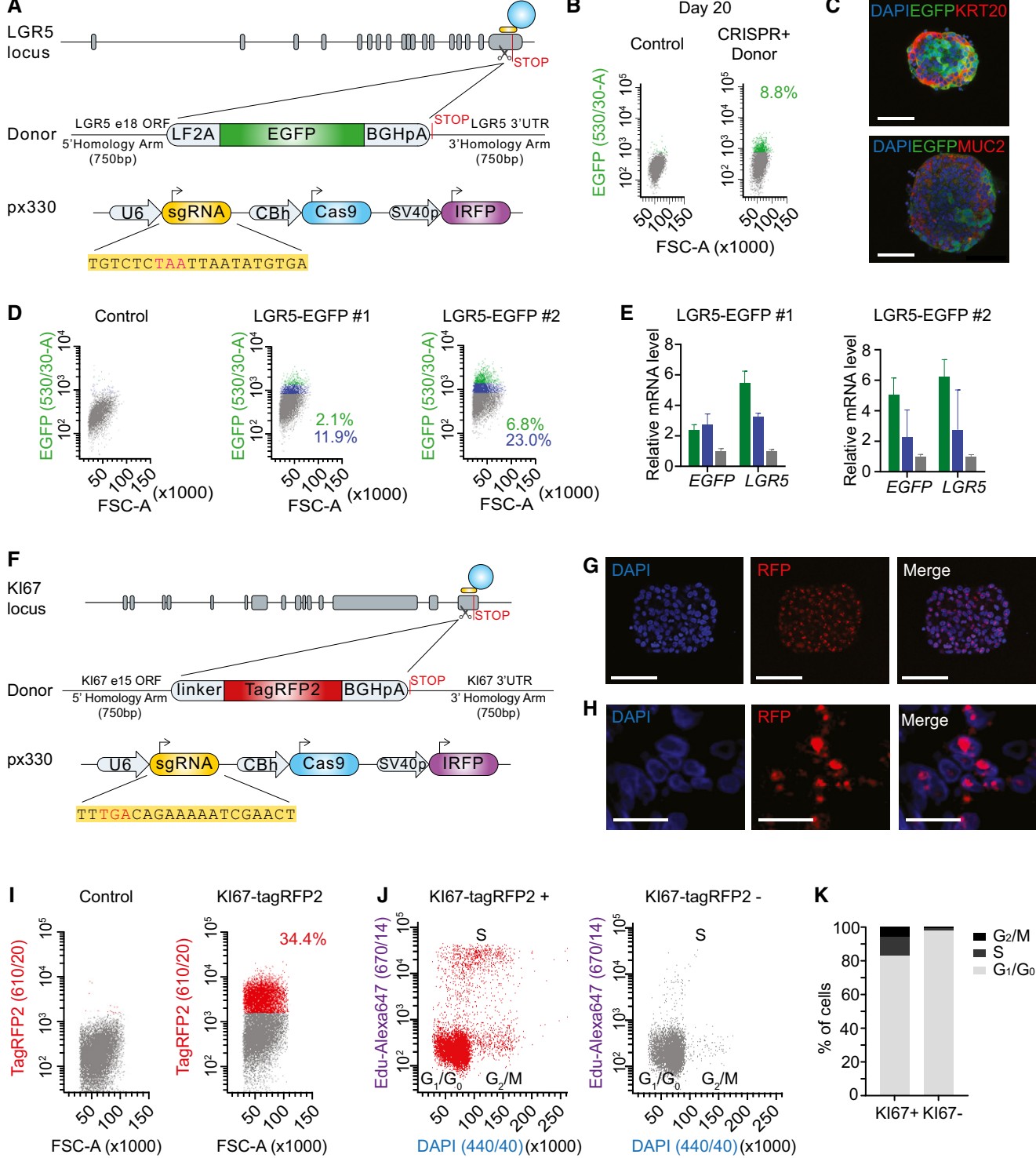

Figure 1.

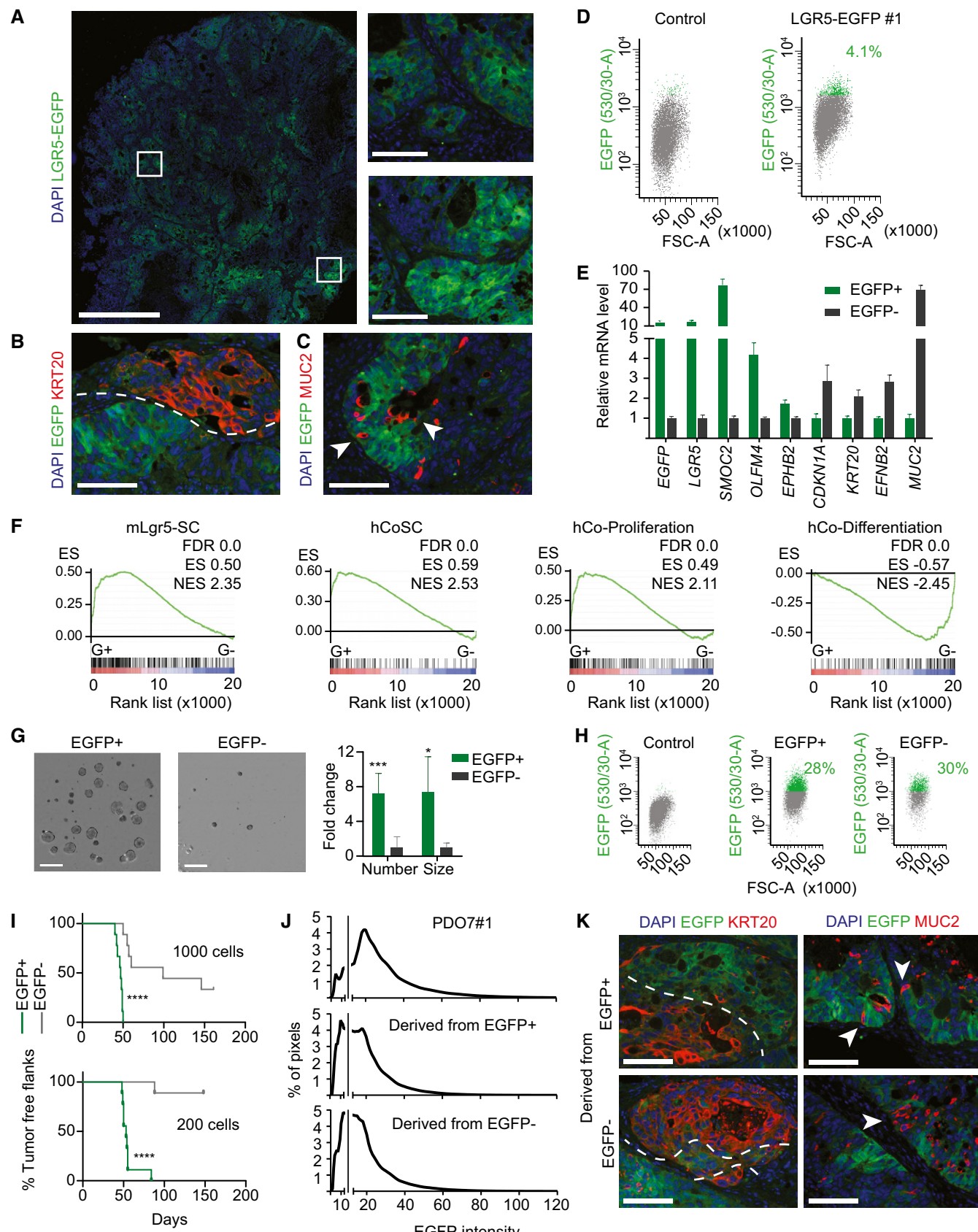

**Figure 2.**

◀

**Figure 2. Characterization of human LGR5⁺ CRC cells *in vivo*.**

A   Representative images of EGFP by immunofluorescence on a section of PDO#7-LGR5-EGFP#1-derived subcutaneous xenograft. White squares indicate the position of the insets. Scale bars indicate 1 mm for the whole xenograft and 100 μm for the insets.

B   Dual immunofluorescence on paraffin sections for KRT20 and LGR5-EGFP showing complementary expression domains. Dashed line delimits expression domain in adjacent glands. Scale bar indicates 100 μm.

C   Dual immunofluorescence on paraffin sections of clone #1 for MUC2 and LGR5-EGFP. White arrows point to LGR5-EGFP⁻/MUC2⁺ cells. Scale bar indicates 100 μm.

D   Representative FACS profiles of EGFP⁺ and EGFP⁻ in EPCAM⁺/DAPI⁻ subpopulation from disaggregated xenografts.

E   Relative mRNA expression level of intestinal stem and differentiation genes for the sorted EGFP⁺ and EGFP⁻ populations. Values show mean ± standard deviation (s.d.) of three measurements.

F   GSEA comparing the expression of signatures of mouse LGR5⁺ cells (MmLgr5-SC), human colon stem cells (hCoSCs), differentiated cells (hCo differentiation), or proliferative crypt cells (Jung *et al*, 2011) in profiled LGR5-EGFP⁺ versus LGR5-EGFP⁻ cells.

G   Representative images and quantification of organoid formation assays from cells purified from PDO#7-LGR5-EGFP#1-derived subcutaneous xenograft (*n* = 4 wells per condition). Data is represented as mean ± s.d. Scale bars indicate 1 mm.

H   Representative flow cytometry analysis of 15-day grown organoids from the EGFP⁺ and EGFP⁻ sorted populations.

I   *In vivo* tumor initiation capacity of 1,000 and 200 sorted cells from PDO#7-LGR5-EGFP#1-derived subcutaneous xenografts. Graphs show Kaplan–Meier plots (*n* = 9 xenografts).

J   Distribution of the EGFP staining intensity in PDO7#1 and in xenografts derived from EGFP⁺ and EGFP⁻ cells. Gray line indicates background fluorescence levels.

K   Dual immunofluorescence for KRT20/EGFP and MUC2/EGFP on paraffin sections of xenografts generated by EGFP⁺ and EGFP⁻ sorted populations respectively. Dashed lines mark stem-like and differentiated-like compartments. White arrows point to secretory cells intermingled in the LGR5⁻ compartment. Scale bars indicate 100 μm.

Data information: Differences in organoid formation assay were assessed with Student's *t*-test and in tumor initiation assay by log-rank (Mantel–Cox) test: *$P$-value < 0.05, ***$P$-value < 0.005, ****$P$-value < 0.0001. The exact $P$-values are specified in Appendix Table S5.

LGR5-EGFP⁻ CRC cells (Fig 2F). We next assessed the clonogenic potential of LGR5-EGFP CRC cells. LGR5-EGFP⁺ cells purified from xenografts displayed several fold higher organoid forming capacity than LGR5-EGFP⁻ cells (Figs 2G and EV3F). Organoids generated by LGR5-EGFP⁺ cells contained both EGFP⁺ and EGFP⁻ tumor cells in a proportion similar to that of the PDO of origin (Figs 2H and EV3G). We obtained similar results using LGR5-EGFP knock-in cells generated from PDO#6 (Fig EV1H–K).

Finally, to assess the capacity of these tumor cell populations to propagate the disease to mice, we inoculated 200 or 1,000 LGR5-EGFP⁺ or LGR5-EGFP⁻ epithelial tumor cells isolated from xenografts into secondary hosts. These experiments showed that the EGFP⁺ cell population was largely enriched in tumor-initiating cells compared to their differentiated EGFP⁻ counterparts (Figs 2I and EV3H). Tumors generated by LGR5-EGFP⁺ cells were populated with stem-like (EGFP⁺/KRT20⁻) and differentiated-like (EGFP⁻/KRT20⁺) tumor cells in similar proportions than the primary xenografts from which they were purified (Fig 2K) thus implying that LGR5-expressing CRC cells undergo self-renewal and differentiation during tumor expansion. Of note, xenografts generated by LGR5-EGFP⁻ cell population were also formed by EGFP⁺ and EGFP⁻ cells with equivalent intensities and proportions to those observed in xenografts derived from LGR5-EGFP⁺ cells (Fig 2J). The expression pattern of the differentiation markers KRT20 and MUC2 was also similar in xenografts arising from the two cell populations (Fig 2K).

## Lineage tracing of human LGR5⁺ CRC cells

As discussed in the introduction, currently it is not possible to perform cell fate-mapping experiments in human cancers similar to those performed in mouse models. To overcome this limitation, we used CRISPR/Cas9 to engineer PDOs containing a lineage-tracing system. We first introduced a Cre recombinase-inducible reporter into the neutral AAVS1 locus (Fig 3A). This reporter consisted of a constitutive Ubiquitin C (UBC) promoter driving the expression of blue fluorescent protein mTagBFP2. This cassette was flanked by LoxP sites so that expression of a downstream tdTomato (TOM) reporter remains blocked until the mTagBFP2 cassette is excised by

Cre recombinase activity. Following the approach described for LGR5-EGFP targeting, we selected long-term mTagBFP2-expressing cells after nucleofection and expanded single cell-derived organoids. Subsequently, we generated a second genomic edition consisting in an LF2A-CreERT2 cassette recombined upstream of the LGR5 stop codon (Fig 3B). The frequencies of correct integrations for these cassettes were 47.8 and 1.78%, respectively (Appendix Table S2). We further confirmed correct integrations of these constructs by PCR as well as by Sanger sequencing of genomic DNA. To test their functionality, we induced PDOs with 4-hydroxytamoxifen (4-OHT) *in vitro*, which demonstrated conversion of mTagBFP2⁺ cells into TOM⁺ cells (Figs 3C and D, and EV4A). We also tested the utility of these constructs *in vivo* by inoculating double-edited PDOs in mice. Analysis of xenografts 96 h after induction with tamoxifen revealed the appearance of a TOM⁺ side population, which retained expression of LGR5 mRNA (Fig EV4B and C) supporting tracing from the LGR5⁺ cell population. In contrast, we did not observe TOM⁺ cells in xenografts growing in untreated mice. Based on a frequency of about 2–4% LGR5-EGFP-hi cells in xenografts (Figs 2D and EV3D), and on the number of TOM⁺ cells arising 96 h post-tamoxifen administration (Fig EV4B), we roughly estimated that recombination occurred in 1 in every 10–20 LGR5-EGFP⁺ cells.

Next, we mapped the fate of LGR5⁺ CRC cells over an extended period of time. The experimental setup is described in Fig 3E. In brief, a cohort of mice bearing edited PDO-derived xenografts were given tamoxifen once tumors were palpable. Mice were sacrificed and tumors analyzed at the indicated time points over 28 days. After this period, tumor pieces were transplanted into secondary recipient mice and xenografts were analyzed for further 4 weeks. We already observed the emergence of TOM⁺ individual cells scattered throughout the tumor glands 96 h after tamoxifen induction (Fig 3F). About 75% of these marks corresponded to isolated cells and the rest to two cell clones (Fig 3G) implying that these experimental conditions enable tracing from individual tumor cells. Quantification of clone size revealed heterogeneity in the growth dynamics of LGR5⁺ CRC cells. Whereas some clones expanded steadily over time, a substantial proportion of LGR5⁺ divided slowly or even remained as individual entities over extended periods (56 days

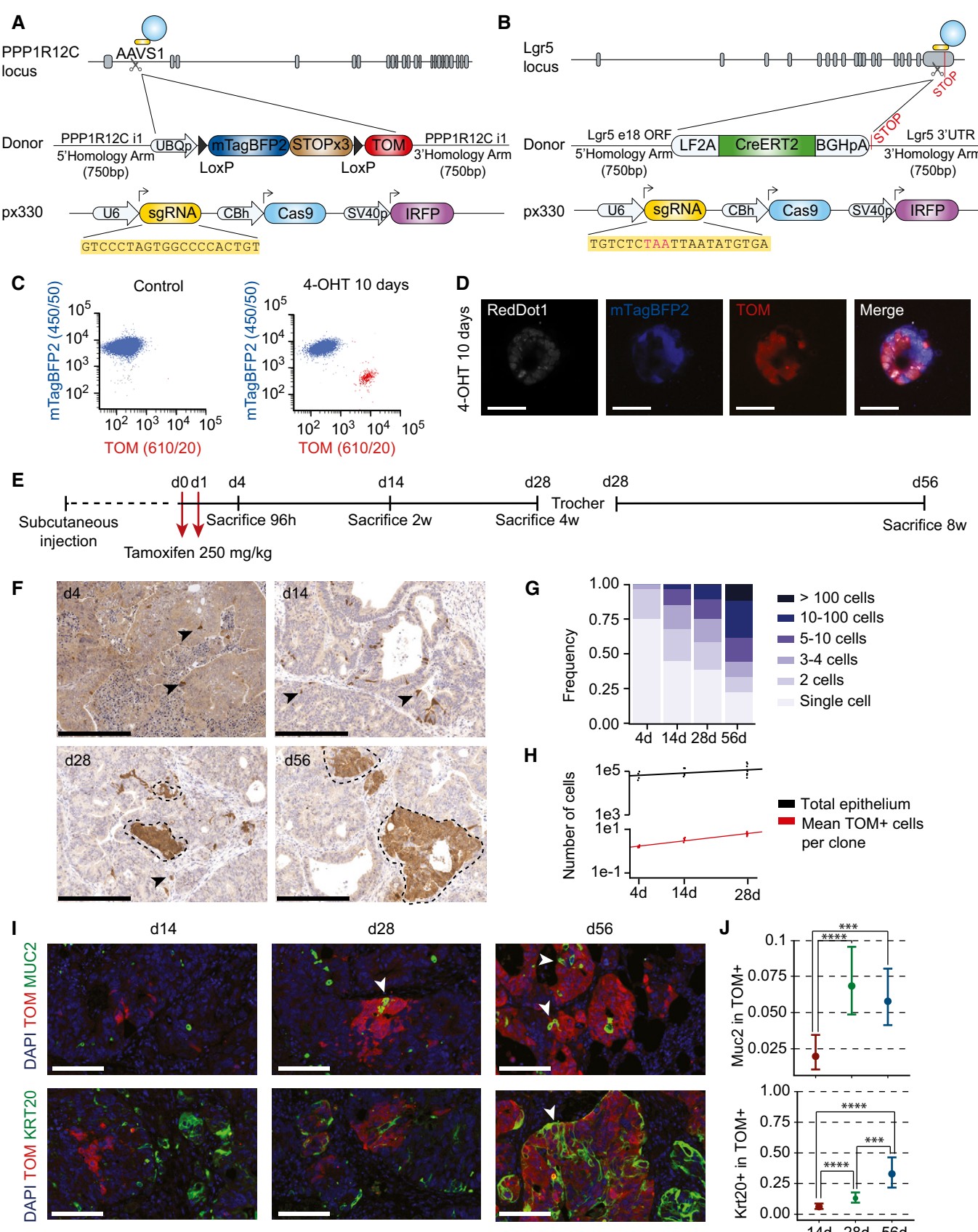

**Figure 3.**

**Figure 3.  Lineage tracing of LGR5⁺ CRCs in human colorectal xenografts.**

A   Design of the donor vector containing lineage-tracing cassette and AAVS1 homology arms.

B   Design of LGR5-CreERT2 donor and CRISPR/Cas9 sgRNA vectors.

C   Flow cytometry analysis of double knock-in PDO#7 carrying AAVS1-LSL-TOM and LGR5-CreERT2 cassettes. Organoids were treated *in vitro* with 1 μM 4-hydroxytamoxifen (4-OHT). About 3.6% of the cells recombined the stop cassette (i.e., expressed low levels of mTagBFP2) and gained expression of TOM.

D   Confocal imaging of double knock-in organoids 10 days after *in vitro* 1 μM 4-OHT addition. Scale bars indicate 50 μm. Note that recombined organoids switch mTagBFP2 to TOM expression.

E   Experimental setup used for lineage-tracing experiments.

F   Representative immunohistochemistry using anti-Tomato antibodies on paraffin sections of the four time points after tamoxifen treatment. Arrowheads point to single and two cell clones. Dashed lines delimit large clones. Scale bars indicate 250 μm.

G   Clone size frequency per time point according to number of cells. Number of clones quantified was 878 for day 4, 2,424 for day 14, 6,940 for day 28, and 6,940 for day 56.

H   Correlation of number of epithelial cells per xenograft and number of cells per clone over time (*n* = 4 xenografts for 4 days time point, *n* = 5 xenografts for 14 days time point, *n* = 8 xenografts for 28 days time point, *n* = 8 xenografts for 56 days time point).

I   Expression domains of TOM and differentiation markers MUC2 and KRT20. White arrowheads indicate double-positive cells. Scale bars indicate 100 μm.

J   Quantification of the number of MUC2⁺ and KRT20⁺ cells within TOM⁺ clones at each time point. Data is represented as the 95% confidence intervals of the measurements. Number of clones assessed was 872 (4 days), 372 (day 14), and 69 (day 28) for KRT20 and 387 (day 4), 611 (day 14), and 130 (day 28) for MUC2. The *P*-value was calculated using a generalized linear model with binomial response. ***P*-value < 0.005, ****P*-value < 0.0001. The exact *P*-values are specified in Appendix Table S5.

in these experiments) (Fig 3F and G). 3D reconstruction from multiple serial tissue sections confirmed the existence of many isolated 1–4 cell clones at day 28 after tamoxifen induction (Fig EV5A–D and Movie EV1). Quantification of clone number over time showed that the number of cells generated by LGR5⁺ cells was directly proportional to the expansion kinetics of the tumor epithelial compartment (Fig 3H). The scaling pattern of the LGR5⁺ cell output is compatible with the hypothesis that tumor growth is largely the result of LGR5⁺ cell activity. Of note, we observed few MUC2⁺ and KRT20⁺ cells in clones during the first 2 weeks of tracing, whereas the frequency of differentiated cells increased after this period (Fig 3I and J). Therefore, in CRC, the progeny of LGR5⁺ cells remains undifferentiated during extended periods of time.

### Marking of quiescent LGR5⁺ CRC cells

The observation that a proportion of LGR5⁺ cell in lineage-tracing experiments produced few progeny may reflect a quiescent state. Indeed, we found that about half of LGR5⁺ cells stained negative for KI67 (Fig 4A and B). To further characterize this cell population, we generated LGR5-EGFP PDOs that expressed TagRFP2 fused to endogenous KI67 protein following the approach described in Fig 1. Analysis of xenografts derived from LGR5-EGFP/KI67-TagRFP2

PDOs confirmed that a large proportion of LGR5-EGFP⁺ cells did not express KI67-TagRFP2 (Fig 4C). In independent xenografts and clones, the fraction of LGR5-EGFP⁺/KI67-TagRFP2⁻ ranged from 20 to 50%. LGR5-EGFP⁺/KI67-TagRFP2⁻ cells purified from xenografts displayed cell cycle profiles that indicated arrest in G1/G0 phase (Fig 4D). Using FACS, we purified KI67-TagRFP2⁺ (K⁺) and KI67-TagRFP2⁻ (K⁻) cells within the LGR5-EGFP⁺ (L⁺) and LGR5-EGFP⁻ (L⁻) gates and compared their gene expression profiles. The L⁻/K⁻ cell population showed downregulation of proliferation genes, upregulation of the cell cycle inhibitor CDKN1A, and expression of markers of terminal differentiation KRT20 implying that they represent mature differentiated CRC cells (Fig 4E). L⁻/K⁺ cells displayed low levels of ISC marker genes and upregulated genes characteristic of early absorptive differentiation such as FABP1 and SI (Fig 4E). By analogy with the normal intestinal epithelium, we hypothesize that this cell population resembles proliferative progenitors undergoing differentiation toward an enterocyte-like phenotype. Our analysis also showed that the L⁺/K⁻ cell population was characterized by downregulation of key genes involved in proliferation and cell cycle progression such as KI67, AURKB, FOXM1, and UBE2C (Fig 4E and F) but retained elevated levels of ISC marker genes including LGR5 and SMOC2 (Fig 4E–G). GSEA further demonstrated an overall downregulation of the proliferative genes in L⁺/K⁻ cells (Fig 4G). A

**Figure 4.   Dual LGR5 and KI67 knock-in PDOs enable separation of quiescent and cycling LGR5⁺ CRC cells.**

A   Representative immunofluorescence image of PDO#7-LGR5-EGFP#1 stained with KI67 antibody. White arrowheads point to double-positive cells; yellow arrowheads point to LGR5⁺/KI67⁻ cells. Scale bars indicate 100 μm.

B   Quantification of KI67⁺ cells within the LGR5⁺ and LGR5⁻ compartments (*n* = 2,749 LGR5⁺ cells, 1,798 LGR5⁻ cells assessed). Data is represented as the 95% confidence intervals of the measurements.

C   Representative FACS profiles from PDO#7-LGR5-EGFP/KI67-TagRFP2 disaggregated xenografts. Only EPCAM⁺/DAPI⁻ cells are shown. The four represented populations are: LGR5-EGFP⁻, KI67-RFP⁻ (gray), LGR5-EGFP⁻, KI67-RFP⁺ (orange), LGR5-EGFP⁺, KI67-RFP⁺ (red) and LGR5-EGFP⁺, KI67-RFP⁻ (green).

D   Cell cycle analysis LGR5-EGFP⁺ and KI67-RFP-positive or KI67-RFP-negative sorted populations. 5,363 and 5,398 cells were analyzed in each case.

E   RT–qPCR analysis of proliferation, stem, and differentiation marker genes in the cell populations defined by EGFP/TagRFP levels. K indicates KI67, and L indicates LGR5. Values show mean ± standard deviation (s.d.) of three measurements.

F   Volcano plot representing gene expression profile of L⁺K⁺ versus L⁺K⁻ purified populations from LGR5-EGFP/KI67-TagRFP2 PDO#7 clone #1. Green dots indicate genes belonging to the human colon stem cell signature, orange dots represent genes belonging to the differentiated cell signature, and blue dots depict genes of the crypt proliferative progenitor signature. Well described genes involved in proliferation are indicated. *P*-values and fold changes computed by fitting a linear model with the R package limma.

G   GSEA comparing LGR5-EGFP⁺ cells positive or negative for KI67-RFP for the signatures used in Fig 2 as well as for signatures derived from mouse crypt LGR5-high/KI67-high or LGR5-high/KI67-low (Basak *et al*, 2014). Note that the only signatures that are differentially expressed between the two populations correspond to proliferative cells derived from either human crypts of from KI67-RFP mice. In contrast, the signature of mouse LGR5-EGFP⁺/KI67-RFP⁻ is significantly enriched in LGR5-EGFP⁺/KI67-RFP⁻ tumor cells.

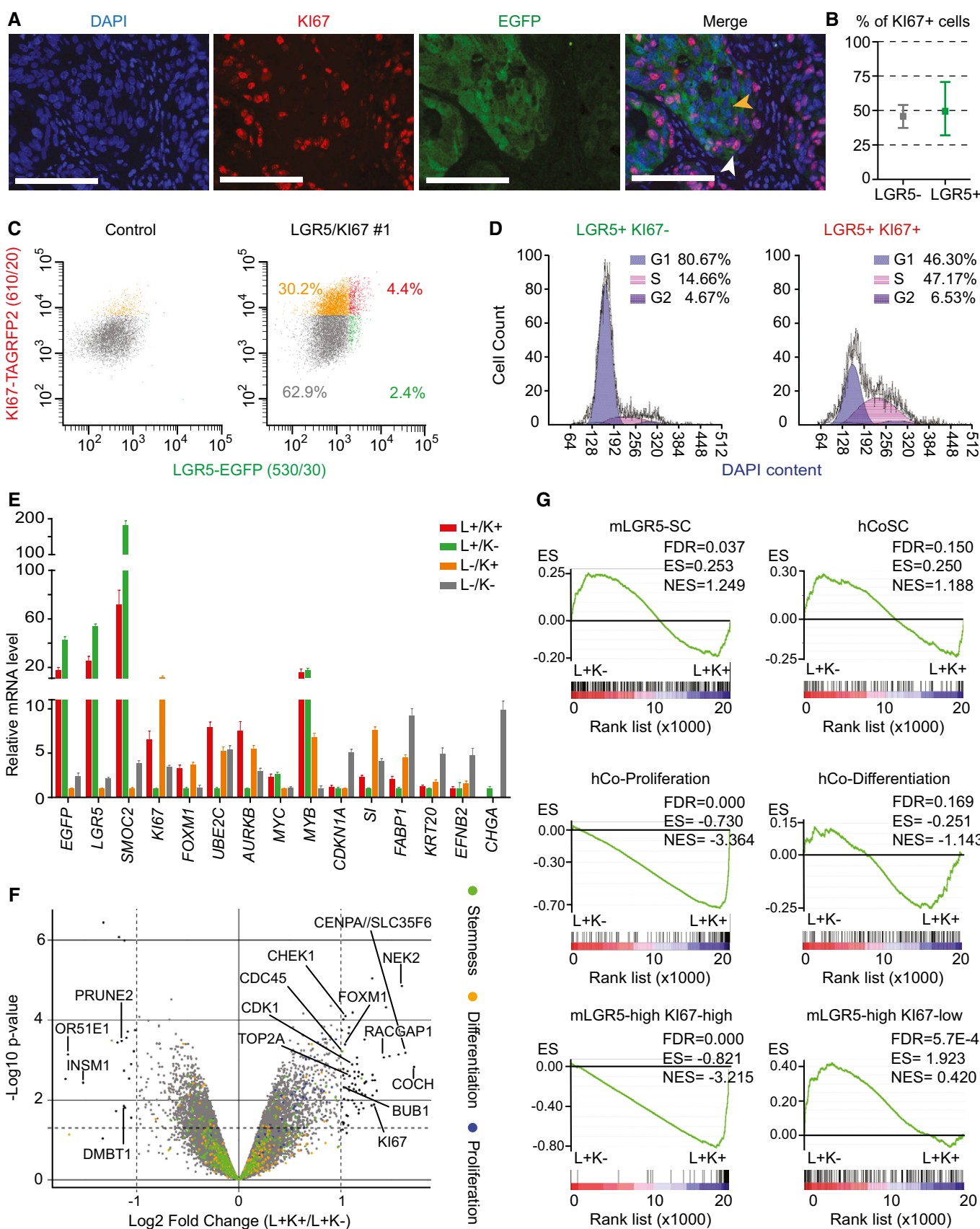

**Figure 4.**

previous work used KI67-RFP knock-in mice to show that a small subset of LGR5$^+$ cells in the healthy mucosa downregulate the expression of KI67 (Basak *et al*, 2014). We used transcriptomic datasets from these mice to identify genes up- and downregulated in normal LGR5$^+$/KI67$^+$ and LGR5$^+$/KI67$^-$ crypt cells (Basak *et al*, 2014). Our analyses showed striking enrichment of these gene sets in L$^+$/K$^+$ versus L$^+$/K$^-$ CRC cells (Fig 4G). In contrast, the oncogenes MYC and MYB, the expression of which is driven by the WNT pathway in CRC (van de Wetering *et al*, 2002), remained elevated in quiescent and proliferating LGR5$^+$ tumor cell populations (Fig 4E).

## Discussion

The combination of organoid and CRISPR/Cas9 technology described herein opens up the study of human tumors through genetic approaches that had only been feasible in animal models. This advance is particularly well suited to analyze phenotypic diversity of cell populations within cancers as it enables labeling and tracing of distinct tumor cells through specific marker genes, which are not necessarily expressed at the cell surface. In contrast, its utility to study genomic heterogeneity is limited, as the current method requires cloning of individual tumor cells to guarantee the accuracy of the genomic insertions. Therefore, tumors generated from edited organoids reflect the behavior of a single stem cell lineage in a genetically homogenous mutational background. To ensure that edited organoids are good surrogates of the parental population, we selected those displaying mutational profiles that overlapped with that of the organoid of origin. Still, although unlikely, we cannot rule out that the few private mutations identified in these monoclonal organoids or other epi-genetic alterations may confer differential properties. Despite these caveats, the possibility of performing cell fate-mapping experiments in human cancers represents a substantial advance. For the first time, this approach enables the analysis of cell lineage relationships in intact tumors and will help address how distinct cell populations contribute to growth, dissemination and resistance to therapy.

Colorectal cancer stem cells had been previously isolated from patient samples using distinct surface markers including CD44, CD133, or EPHB2, which enrich in populations of tumor-initiating cells (O'Brien *et al*, 2007; Dalerba *et al*, 2007; Ricci-Vitiani *et al*, 2007; Merlos-Suarez *et al*, 2011). In normal colonic mucosa, these markers are expressed broadly throughout the stem and transient amplifying compartments (Zeilstra *et al*, 2008; Snippert *et al*, 2009; Jung *et al*, 2011). In contrast, the expression domain of LGR5 is restricted to ISCs (Barker *et al*, 2007) yet the analysis of LGR5-expressing cells in human CRCs had not been possible due to the lack of good reagents. Our work shows that human LGR5$^+$ CRC cells express the gene program of normal ISCs, are clonogenic *ex vivo*, and display robust tumor-initiating capacity in xenograft assays. We also performed for first time experiments of lineage tracing in human CRC, which demonstrate that LGR5$^+$ tumor cells produce progeny over long periods of time, which undergo differentiation to distinct lineages. Hence, our work reinforces the notion that despite the accumulation of multiple genetic alterations, human CRCs are governed by a cell hierarchy reminiscent of that present in the normal intestinal epithelium. Our observations revealed two other interesting aspects. First, the kinetics of differentiation of tumor cells in CRC appears to be a relatively slow process compared to the

normal epithelium, where the progeny of LGR5$^+$ ISCs undergoes differentiation 2–3 days after they leave the crypt base (Clevers, 2013). In contrast, clones produced by LGR5$^+$ CRC cells were largely devoid of differentiated cells, which only started to accumulate after 2 weeks approximately. This delayed differentiation fits in well with the observation that LGR5$^+$ and KRT20$^+$ tumor cells reside in complementary compartments rather than intermingled in the same area and may suggest that distinct tumor niches facilitate stem or differentiation states. Second, whereas the vast majority of normal ISCs remain in a proliferative state (Schepers *et al*, 2012; Basak *et al*, 2014), a substantial proportion of LGR5$^+$ CRC cells contribute with few progeny according to the lineage-tracing data. This subset of inactive LGR5$^+$ cells likely represent LGR5$^+$/KI67$^-$ cells identified in double-reporter knock-in PDOs. These data further support previous clonal analysis of CRC using lentiviral marking of patient samples, which revealed the existence of dormant cells that can be reactivated upon passaging or chemotherapeutic treatment (Dieter *et al*, 2011; Kreso *et al*, 2013). Finally, the finding that the progeny of LGR5$^+$ tumor cells scales with the total number of epithelial cells fits in well with the hypothesis that CRC growth is the result of the activity of multiple LGR5$^+$ tumor stem cells. Nevertheless, our data does not rule out that LGR5$^-$ cells could contribute equally to tumor growth. In the normal intestinal epithelium, differentiated cells can opportunistically replace LGR5 + ISCs through plasticity (van Es *et al*, 2012; Tetteh *et al*, 2016), implying that the ISC phenotype is not hardwired but rather is induced by the niche. Thus, it is likely that LGR5$^+$ and LGR5$^-$ tumor phenotypes are also plastic. Our observation that the xenografts generated by LGR5$^-$ cells display cellular patterns equivalent to those produced by LGR5$^+$ cells may indicate interconversion of the two cell populations in these transplantation assays, yet confounding effects such as suboptimal isolation of the LGR5-EGFP$^-$ population could as well explain our results. Proper assessment of cell plasticity will require mapping the fate of LGR5$^-$ cells in intact tumors through genetic strategies equivalent to those described herein.

## Materials and Methods

### Organoid cultures

PDO#6 and PDO#7 have been previously described (Calon *et al*, 2015). In brief, the tumor sample used to expand PDO#6 was obtained from an individual treated at Hospital de la Santa Creu i Sant Pau, under informed consent and approval of the Tumor Bank Committees according to Spanish ethical regulations. The study followed the guidelines of the Declaration of Helsinki, and patient identity for pathological specimens remained anonymous in the context of this study. Tumor cells were grown as organoids embedded in BME2 (basement membrane extract 2, AMSbio) using a modification of the media described by the Clevers laboratory (van de Wetering *et al*, 2015) (Advanced DMEM/F12, 10 mM HEPES, 1× Glutamax; 1× B-27 without retinoic acid, 20 ng/ml bFGF (basic fibroblast growth factor); 50 ng/ml EGF (epidermal growth factor), 1 µM LY2157299, 10 µM Y27632, and recombinant Noggin (100 ng/ml). PDO#7, a kind gift from G. Stassi (University of Palermo), was obtained from the dissociation of whole CRCs in suspension as described elsewhere (Lombardo *et al*, 2011). Upon arrival to our laboratory, they were cultured with

the medium described above. All cells were tested weekly for mycoplasma contamination with negative results.

## Xenograft assays

All experiments with mouse models were approved by the Animal Care and Use Committee of Barcelona Science Park (CEEA-PCB) and the Catalan government. We inoculated 150,000 cells (PDO#7) or 2 million cells (PDO#6) in a format of 5- to 7-day grown organoids subcutaneously into NOD/SCID female mice in 50% BME2-HBSS. Generally, a maximum of 4 xenografts were generated per animal. Tumor volume was measured with manual calipers. For tumor initiation assays, viable single human cells (EPCAM-positive; DAPI-negative) from disaggregated xenografts were sorted according to their EGFP levels and subsequently transplanted into recipient mice in 100 μl of BME2:HBSS (1:1).

## Lineage tracing and clonal analysis

Cohorts of NOD/SCID mice were inoculated with organoids as described above. When tumors were palpable, mice were given two consecutive doses of tamoxifen (250 mg/kg) to maximize recombination. Mice were sacrificed at indicated time points and tumors were processed for histological analysis. Clone sizes over time were determined in histological sections and scored by manual counting or image analysis software. We averaged measures from distinct sections and xenografts at each time point. Size of clones present at 4-day post-tamoxifen induction was assessed manually. For subsequent time points, we analyzed images using Interactive Learning and Segmentation Toolkit, Ilastik software (www.ilastik.org). We set the algorithm parameters so that adjacent clones or cells that were not in contact computed as independent clones. A full description of the methodology used for clonal analysis is included in the Appendix Supplementary Methods.

## Transcriptomic profiling

RNA from selected tumor cell populations isolated from xenografts by FACS (1,000–5,000 cells per sample) was amplified using pico-profiling (Gonzalez-Roca *et al*, 2010) and subsequently hybridized on Primeview arrays (Affymetrix). Gene expression was analyzed using standard methodology as described in the Appendix Supplementary Methods. Data have been deposited at Gene Expression Omnibus (GSE92960 and GSE92961).

A detailed description of the methods is included in the Appendix Supplementary Methods.

**Expanded View** for this article is available online.

## Acknowledgements

We thank all members of the Batlle laboratory for support and discussions. We are grateful for the excellent assistance by IRB Barcelona core facilities for histology, functional genomics, biostatistics/bioinformatics, and advanced digital microscopy, as well as the flow cytometry and animal facilities of the UB/PCB. This work has been financed by the European Research Council (ERC advanced grant 340176). Work in the laboratory of EB is also supported by Fundación Botín and Banco Santander, through Santander Universities. IRB Barcelona is the recipient of a Severo Ochoa Award of Excellence from the MINECO.

## The paper explained

### Problem
The study of stem cell hierarchies and of other sources of cellular diversity in human cancers has been largely based on experiments of tumor cell isolation from dissociated patient samples. These experiments impose a number of limitations; first, the requirement of antibodies against specific membrane proteins to label particular cell populations limits the repertoire of cell phenotypes that can be analyzed. Second, the necessity to dissociate the sample impedes the examination of tumor cell populations in an intact environment (i.e., in a growing tumor).

### Results
We combine two novel methodologies—patient-derived tumor organoids (PDOs) and genome editing tools—and apply them to study the cellular heterogeneity of CRC without the limitations described above. To illustrate the utility of this approach, we studied LGR5[+] cells in human CRC, the analysis of which has been hampered by lack of good commercial reagents to recognize this protein. Using CRISPR-mediated targeting, we engineered PDOs carrying an EGFP reporter cassette recombined at LGR5 locus. We discovered that the LGR5[+] tumor cell population expresses a gene program similar to that of normal ISCs. In xenograft experiments, human LGR5[+] tumor cells propagated the disease to recipient mice with high efficiency implying that this cell population is largely enriched in tumor-initiating cells. Furthermore, we generated PDOs bearing a lineage-tracing cassette and subsequently mapped the fate of LGR5[+] cells in intact tumors. We found that LGR5[+] cells display long-term self-renewal and multilineage differentiation capacity. Finally, by generating dual LGR5-EGFP/Ki67-RFP knock-in PDOs, we characterized a population of quiescent stem cell-like cells in human CRCs.

### Impact
The approach described herein has broad applicability to analyze the phenotypic diversity of human tumors. In essence, it brings the power of mouse genetic tools to study human cancer. Among many other possibilities, it enables lineage-tracing experiments that can help to elucidate the behavior of distinct tumor cell populations during growth, dissemination, and resistance to therapy.

## Author contributions

EB conceived the study and wrote the manuscript. CC and GT designed experimental work, executed experiments, and helped with manuscript writing. DS provided technical support with targeting vector generation and genome editing of organoids. XH-M performed mice work. AM-S and ES helped designing and conceptualizing the study. MS provided crucial help with immunohistochemistry and organoid cultures. CS-OA performed all statistical analyses. MA and ST contributed to image analysis and 3D reconstruction of serial sections.

## Conflict of interest
The authors declare that they have no conflict of interest.

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
