## [Review Process File · EMBO Molecular Medicine]

A genome editing approach to study cancer stem cells in human tumors

Carme Cortina, Gemma Turon, Diana Stork, Xavier Hernando-Momblona, Marta Sevillano, Mònica Aguilera, Sébastien Tosi, Anna Merlos-Suarez, Camille Stephan-Otto Attolini, Elena Sancho and Eduard Batlle

Corresponding author: Eduard Batlle, IRB Barcelona

Review timeline:

Submission date:	09 January 2017
Editorial Decision:	16 February 2017
Revision received:	30 March 2017
Editorial Decision:	07 April 2017
Revision received:	07 April 2017
Accepted:	11 April 2017

Transaction Report:

Editor: Roberto Buccione

1st Editorial Decision

16 February 2017

Thank you for the submission of your manuscript to EMBO Molecular Medicine. We are sorry that it has taken longer than usual to get back to you on your manuscript. In this case we experienced some difficulties in securing three appropriate expert reviewers, and then obtaining their evaluations in a timely manner. Finally, we also wished to discuss the evaluations further.

As you will see, the reviewers essentially paint a positive picture accompanied by the expression of a number of partially overlapping concerns, mostly focused on interpretation but not questioning the validity, quality and usefulness of the work.

Many comments are geared towards increasing the overall medical relevance and potential clinical consequences of the conclusions. Ultimately, the main issue that all reviewers mention is that there may be more plasticity in the system than the is given credit for, in the sense that Lgr5⁻ cells can give rise to Lgr5⁺ ones. This was confirmed during our reviewer cross-commenting exercise.

In conclusion, while publication of the paper cannot be considered at this stage, we would be pleased to consider a revised submission, with the understanding that the Reviewers' concerns must be addressed including with additional experimental data where appropriate and that acceptance of the manuscript will entail a second round of review.

Specifically, we would like you to especially focus on the plasticity concern, perhaps by better characterizing the organoids and xenografts from the Lgr5⁻ population. Please note however, that we agree with the reviewers that while this further experimentation would go a long way to increase the

overall solidity and relevance of your findings, it will not ultimately impact on the overall favorable opinion on the manuscript as long as you carefully work on clarifying and improving the manuscript including in terms of interpretation, as suggested by the reviewers.

I look forward to seeing a revised form of your manuscript as soon as possible.

***** Reviewer's comments *****

Referee #1 (Comments on Novelty/Model System):

Using an elegant CRSIPR/Cas9 gene editing strategy the authors demonstrate profound cellular heterogeneity within CRC and provide compelling evidence that LGR+ cells are dominant in tumor progression and generate phenotypically and functionally diverse cells in vivo. Such an approach has not been done before in primary human cancer tissue. The study is of high relevance as the cell fate mapping strategy described may allow for a broader applicability in different human cancers.

Referee #1 (Remarks):

Cortina et al report a highly interesting cell fate tracing study in human CRC xenografts and organoid cultures. Using an elegant CRSIPR/Cas9 gene editing strategy they demonstrate profound cellular heterogeneity within CRC and provide compelling evidence that LGR+ cells are dominant in tumor progression and generate phenotypically and functionally diverse cells in vivo. The study is of high relevance as the cell fate mapping strategy described may allow for a broader applicability in different human cancers. The experimental results are clearly described and discussed. I have some minor suggestions on how to improve the manuscript:

Minor points:

- Clear evidence is provided that LGR5+ cells are more tumorigenic than LGR5- and generate LGR5+ cells and cells that lose LGR5 expression but gain some markers that are associated with differentiation in normal intestinal epithelium. It would be very interesting to understand whether this phenotypic differentiation is unidirectional or whether LGR5- cells can regenerate LGR5+ cells after transplantation. What was the proportion of LGR5+ and LGR5- cells in xenografts generated by LGR5- CRC cells and do LGR5+ cells in such tumors regain full tumorigenic potential?
- Affiliation 3 has no reference in the author listing.

Referee #2 (Comments on Novelty/Model System):

The details are in the report to the authors.

I evaluated the medical as "medium" because there are no DIRECT applications to the clinics. However, in terms of conceptual medical impact, I would rank it as "high".

Referee #2 (Remarks):

In this study, Cortina and collaborators report the use of the CRISPR-Cas9 system to label specific cell populations in CRC patient-derived organoids (PDOs) and their analysis in vitro and in vivo. This was applied to cell populations expressing the LGR5 putative cancer stem cell marker and the KI67 proliferation marker. The strategy used is generally well explained in the results section (but see comment below).

LGR5-GFP PDOs were grafted into nude mice. GFP was detected in some of the cells, which were characterized in terms of their differentiation status, gene expression profiles and clonogenicity (in vitro and in vivo). The authors concluded that LGR5-GFP+ cells have properties similar to those of mouse Lgr5+ stem cells.

The authors then generated additional PDO lines for lineage tracing experiments. Clones of cells originating from LGR5-expressing tumor cells were detected and such clones were found to include different cell types. Furthermore, many LGR5-expressing cells did not expand as clones but remained as single isolated cells. Further analyses demonstrated that these cells do not express the

KI67 proliferation marker. Finally, gene expression profiling of the four cell populations characterized by LGR5 and/or KI67 defined physiologically distinct cell populations, including two that expressed stem cell markers, and both KI67-positive and negative cells.

Comments:

The description of the gene editing strategy contains an ambiguity: the authors indicate they sorted cells that incorporated the donor vector (IRFP+). According to figure 1A, the donor construct does not contain the IRFP reporter. Do the authors postulate that both px330 and donor vectors are co-transfected? Please clarify. In addition, the methods to genetically edit the PDOs are not explained in the methods section, despite the statement to the contrary "The targeting strategy is summarized in Figure 1A and detailed in methods sections." Given that this methodological breakthrough represents one of the most important and novel aspects of the paper, this method must be explained in detail.

In figure 2A, the PDO line-of-origin of the clone (named clone #1) used for the characterization is not clear. In addition, the characterization of LGR5+ cells should be detailed further for the second (#7) PDO (instead of only stating that the results were validated in a second PDO line). In particular, details should be given about the proportion of cells that are GFP+ and their gene expression profile. Extending this analysis to additional PDOs, representative of distinct classes of colorectal cancer, would greatly strengthen the manuscript.

In figure 2H, I understand that organoids generated from LGR5-GFP- cells contain a number of GFP+ cells similar to that of organoids from LGR5-GFP+ cells. This should be described in the text and commented upon in terms of implication for the nature of cancer stem cells and their biology. This comment also applies to figure 2I.

In figure 3D, why is one of the two organoids not recombined? Does this reflect a different genotype, low recombination efficiency or LGR5-independent organoid growth (this could be easily tested by double staining ER and TOM)?

In the same paragraph, I don't understand the sentence "Analysis of xenografts 96h after induction with tamoxifen revealed the appearance of a TOM+ side population, which expressed LGR5 mRNA (Supplementary Fig 4) supporting tracing from the LGR5+ cell population". If TOM+ cells express the LGR5 mRNA, this supports the presence of LGR5+ cells in the xenograft rather than tracing from these cells (which should produce a majority of LGR5-, differentiated cells). Please correct or clarify. By the way, what proportion of the total cells is this category? Is it similar to that of LGR5-GFP+ cells?

In figure 3H, I agree that the results are compatible (i.e. do not contradict) with the hypothesis that LGR5 activity sustains the tumor growth. However, an experiment with a PDO line engineered to delete LGR5+ cells (for instance diphtheria toxin-based) would be much more convincing. In its absence, the sentence needs to be reworded.

In figure 3J, the dynamics of TOM+ cell differentiation is confusing. What are the cells produced from LGR5+ cells that do not display differentiation markers? Do they still express LGR5? This may result in much higher proportions of LGR5-GFP+ cells in the reporter experiments.

In figure 4C, it seems to me that two thirds of the LGR5-GFP cells express KI67-RFP. Why do the authors state that "a large proportion of LGR5-GFP+ population did not express KI67-tagRFP2"? If this is because the authors anticipated more KI67-expressing LGR5-GFP cells, based on their knowledge of the mouse stem cell population, this should be explained.

The kinetics of tumor cell turnover seems slower than in the healthy tissue. As this contrasts with some previous reports, this should be discussed in more detail. Alternatively, perhaps the authors could estimate the efficiency of Cre-mediated recombination within tumors?

In summary, although this manuscript needs a few improvements, the two major breakthroughs presented, i.e. (1) the technology for genome editing of human CRC cells and (2) the function of LGR5 within human CRC cells and the identification of a LGR5+;KI67- tumor cell population, make an important contribution in the field.

Referee #3 (Comments on Novelty/Model System):

The data here are interesting and novel and the statistics are fine. It is interpretation that is at issue currently.

Referee #3 (Remarks):

This is an interesting paper that describes development of methodology to test functionally "cancer stem cell" properties using cutting-edge techniques involving patient-derived organoids, CRISPR/Cas9 induced lineage tracing and cell cycle probes, and sequential xenografting. In all, it is an interesting set of methodologies, with some interesting findings. There are some key overinterpretations and lack of interpretations that should be rectified. There are a number of other smaller deficiencies. More or less in order of import:

1) The authors make comparisons of CRC stem cell hierarchy and kinetics to that of normal cells: "Clones generated by LGR5+ cells contained MUC2+ and KRT20+ cells (Fig 3I) yet we found that the frequency of differentiated cells in each clone increased over time (Fig 3J). Therefore, loss of the stem cell program and gain of differentiation traits in CRC is a relatively slow process compared to normal intestinal epithelium. " How do these results compare with normal human colonic epithelium? The authors should cite and describe the kinetic stem/differentiated cell studies of this sort in normal epithelium that have been performed and can be used to compare to the current xenografted CRC studies.

2) It is not clear why the authors think that some LGR5+ cells produced few progeny. First, what is this based on? The fact that there maintains some single and double clones? These could be artifacts of tissue cutting (they may be connected to bigger colonies above or below the plane of section) or it could be simply that the tamoxifen induction is slow and/or leaky with new clones popping up over time. At any rate, it isn't clear that simply being Ki-67- implies any long-term state of quiescence. Or rather, perhaps the authors should define quiescence. There would seem to be several parameters involved: eg, how long G0/G1 and G2 phases are for the average LGR5+ cell. Perhaps that parameter is simply stochastic with broadly distributed normal curve? Another parameter is how long the Ki67-tag and mRNA signal perdures relative to phases of the cell cycle. One could easily envision a scenario, where all LGR5+ cells are part of essentially the same pool, each cell with normally (bell-shaped curve) distributed chance of entering the cell cycle, and there is a certain perdurance of Ki67 such that some of these cells at any time don't label with Ki67. In this scenario, there are no special, separate, quiescent LGR5+ cells, just a certain likelihood at any given time point that some of the population will be caught without detectable Ki67. If you could follow individual cells, perhaps many of these, 24h later would now be Ki-67 positive. The gene expression profiles do not really help with this analysis, as most depicted genes (except Ki67, of course) are relatively similar. Aurkb is decreased in Ki67-, but this is a G2/M transcript. Cdkn1a, on the other hand, is also a proliferation marker, and it is decreased in both Ki67+ and ki67-, but it is a G1-S transition gene. In short, it is not easy to conclude any kind of quiescent population in these Lgr5+/Ki67- cells without understanding the cell cycle kinetics of LGR5 and Ki67. If the authors would really like to make this case of a dedicated quiescent cell population, then studies of colony-forming potential would be useful. Or even simply taking the + and - populations from the organoids and seeing if they have any immediate differences or can interconvert.

3) Similarly, the authors ignore plasticity with Lgr5- cells also being able to become Lgr5+. For example, this seems to be the basis of the claim that because cells that had been Lgr5+ (tomato+) at one point grows at the same rate as the overall population, this means that "tumor growth is largely the result of LGR5+ cell activity." One could certainly envision other interpretations to explain these results. For one, many of the cells in the tumor (LGR5+ and -) might divide symmetrically, and the LGR5+ cells might simply have a rate of division that is faster than the other cells'. To rule this out, strict bookkeeping of LGR5+ clones in terms of which cells are LGR5+ and which not would have to be done (currently not possible with no LGR5 antibodies). In short, why are we to assume KRT20+/LGR5- cells can't themselves divide? And they might also be plastic and become LGR5+ themselves, which may happen in Fig. 2H. I think it's better not to assume too much that differentiation goes only in an LGR5+ to negative direction. That isn't even sure in vivo in normal animals.

4) Similarly, Fig. 2H seems to show that Lgr5-EGFP- cell-derived organoids eventually generate the same proportion of EGFP+ cells. Does that mean EGFP- cells can give rise to EGFP+ cells or that the few LGR5+ cells that were sorted into the GFP- fraction eventually proliferated sufficiently to "catch up" with the EGFP+ population? This is a key question because it would indicate the Lgr5-population is plastic as occurs when Lgr5+ cells are lost in normal tissue (eg following irradiation).

In sum: The paper largely reaches the conclusion that Lgr5+ lineage marks a stem cell population within a tumor. This cell lineage gives rise to a population of differentiated and differentiated-like cell populations within xenografts. If this is the true hierarchy, then why can EGFP-negative cells form organoids AND tumors? The characterization of tumors from such cells appear deficient. This is an interesting finding because one can speculate about other scenarios: plasticity where an EGFP-negative cell can occasionally (or facultatively) be recruited back to be a "stem cell," or alternative hierarchies where multiple "stem cell" lineages exist, or where Lgr5+ cells are an intermediate population, and another cell within the EGFP-negative population is the stem cell. The knock-in allele could also be sporadically silenced, and would need to be verified on single EGFP-negative cells by RT-PCR to determine if they are expressing Lgr5. Again, a plasticity scenario may be plausible in light of data presented in Figure 2H, where tumors derived from EGFP-negative cells were dissociated and plated to form organoids, which then contain EGFP-positive cells. Thus, EGFP-negative cells gave rise to EGFP-positive cells. Discounting this phenomenon (as the authors appear to do) is dangerous if it perpetuates a framework where one might think that therapeutic elimination of Lgr5+ cells would kill the tumor - not true if plasticity between lineages occur. This isn't really surprising with transformed cells, which frequently change phenotypic states (EMT, MET, quiescence, etc.). A discussion of these data and these phenomenon must be included.

5) Also, perhaps the authors should discuss what it means that clone #1 had "few mutations compared to the parental population"? Did 64 de novo mutations occur in culture relative to the original patient isolate? Are the organoids changing that much, constantly? Can these modified PDOs be kept indefinitely? What about the increasing tumor burden implied by the large number of mutations vs. parental strain? These issues must be detailed in a paper whose impact will be determined by how well future users will be able to take advantage of these techniques.

Smaller problems:

Why is there no discrete population of EGFP+ cells in so many FACS plots like fig. 1B,D; 2E,H? The histological assessment in Fig. 1D and 2A-D seems much more as expected with clear positive and negative cells. What do controls that weren't transfected at all look like? In other words, is this a question of background fluorescence or something else? Similarly, the increase in Egfp in qPCR is relatively modest (Fig. 1E). Is this because the EGFP- cells actually have EGFP transcripts that aren't turned over? What Ct levels were these negative cells say vs. non-transfected cells? The lack of decreased protein in "negative" cells might be due to GFP perdurance, which might lead to cells that are negative for mRNA be sorted in the EGFP+ category. That would lead to less differential expression also but of a different sort, where the numerator (EGFP+) cells are actually much lower in EGFP mRNA expression than would be expected and drives down fold increase. What Ct levels were there in EGFP^{Hi} vs. negative vs. non-transfected?

Similarly, there seems to be a great deal of fluorescence in control populations in Fig. 1I, though, of course, the red channel is obviously substantially elevated in Ki67-TagRFP.

P4, lines 19-20: IRFP+ marks the px330 vector (not the donor)

Why do the authors say that some glands are Lgr5+ and KRT20+, which implies that the Lgr5 and KRT glands are separate entities? Wouldn't the authors expect that glands would be mixes of "stem cell"-like cells and the KRT20+ and MUC2+ cells that arise from them? If not, how would KRT20+/LGR5- glands come about?

What is the control in Fig. 4C?

We thank the reviewers for their comments and insights, which have helped improve the manuscript. The aim of our work was to present a new methodology to study tumor cell heterogeneity and exemplify its potential to analyze cancer stem cells. Many of the reviewer's comments relate to the biology of CRC - including the extent of tumor cell plasticity or the role of quiescent tumor cells - rather than to the approach itself. We do agree with the reviewers that these are crucial questions and so we have tried to address them as much as possible or at least reflect the reviewers ideas in the revised version. We, however, think that an in depth analysis of tumor cells dynamics is somewhat beyond the scope of the present work. Additions to the new version are labeled in blue in the main text.

Referee #1 (Comments on Novelty/Model System):

Using an elegant CRSIPR/Cas9 gene editing strategy the authors demonstrate profound cellular heterogeneity within CRC and provide compelling evidence that LGR⁺ cells are dominant in tumor progression and generate phenotypically and functionally diverse cells in vivo. Such an approach has not been done before in primary human cancer tissue. The study is of high relevance as the cell fate mapping strategy described may allow for a broader applicability in different human cancers.

Referee #1 (Remarks):

Cortina et al report a highly interesting cell fate tracing study in human CRC xenografts and organoid cultures. Using an elegant CRSIPR/Cas9 gene editing strategy they demonstrate profound cellular heterogeneity within CRC and provide compelling evidence that LGR⁺ cells are dominant in tumor progression and generate phenotypically and functionally diverse cells in vivo. The study is of high relevance as the cell fate mapping strategy described may allow for a broader applicability in different human cancers.

We thank the reviewer for his/her positive comments.

The experimental results are clearly described and discussed. I have some minor suggestions on how to improve the manuscript:

Minor points:

- Clear evidence is provided that LGR5⁺ cells are more tumorigenic than LGR5⁻ and generate LGR5⁺ cells and cells that lose LGR5 expression but gain some markers that are associated with differentiation in normal intestinal epithelium. It would be very interesting to understand whether this phenotypic differentiation is unidirectional or whether LGR5⁻ cells can regenerate LGR5⁺ cells after transplantation. What was the proportion of LGR5⁺ and LGR5⁻ cells in xenografts generated by LGR5⁻ CRC cells and do LGR5⁺ cells in such tumors regain full tumorigenic potential?

In Figure 2K we now provide evidence that LGR5-GFP⁻ cells generate xenografts that display similar phenotypic diversity than those produced by LGR5⁺ cells, including undifferentiated LGR5-GFP⁺ cells and LGR5-GFP⁻ cells that express KRT20 or MUC2. As suggested by the reviewer, we quantified the number of LGR5-GFP⁺ cells present in xenografts generated by the two tumor cell populations and found equivalent distribution and GFP intensities (Figure 2J- Figure EV2). Whereas these data suggest that some LGR5-GFP⁻ cells were capable of regenerating the phenotypic diversity of the tumor of origin (and therefore that there is some degree of plasticity in the CRC hierarchy), it is not possible to rule out that we had purified a small fraction of LGR5-GFP⁺ cells within the LGR5-GFP⁻ gate. A proper assessment of cell plasticity awaits upcoming experiments of lineage tracing from LGR5-GFP⁻ cell population. We have discussed these ideas in the revised version of the manuscript. Finally, because of the limited time to undertake this revision and the constraints imposed by competing manuscripts, we have not been able to assess the tumorigenic potential of LGR5-GFP⁺ cells isolated from LGR5-GFP⁻ derived xenografts as these experiments will take several months to complete.

- Affiliation 3 has no reference in the author listing. This is now corrected. Thank you.

Referee #2 (Comments on Novelty/Model System):

The details are in the report to the authors.

I evaluated the medical as "medium" because there are no DIRECT applications to the clinics.

However, in terms of conceptual medical impact, I would rank it as "high".

Referee #2 (Remarks):

In this study, Cortina and collaborators report the use of the CRISPR-Cas9 system to label specific cell populations in CRC patient-derived organoids (PDOs) and their analysis *in vitro* and *in vivo*. This was applied to cell populations expressing the LGR5 putative cancer stem cell marker and the KI67 proliferation marker. The strategy used is generally well explained in the results section (but see comment below).

LGR5-GFP PDOs were grafted into nude mice. GFP was detected in some of the cells, which were characterized in terms of their differentiation status, gene expression profiles and clonogenicity (*in vitro* and *in vivo*). The authors concluded that LGR5-GFP+ cells have properties similar to those of mouse Lgr5+ stem cells.

The authors then generated additional PDO lines for lineage tracing experiments. Clones of cells originating from LGR5-expressing tumor cells were detected and such clones were found to include different cell types. Furthermore, many LGR5-expressing cells did not expand as clones but remained as single isolated cells. Further analyses demonstrated that these cells do not express the KI67 proliferation marker. Finally, gene expression profiling of the four cell populations characterized by LGR5 and/or KI67 defined physiologically distinct cell populations, including two that expressed stem cell markers, and both KI67-positive and negative cells.

Comments:

The description of the gene editing strategy contains an ambiguity: the authors indicate they sorted cells that incorporated the donor vector (IRFP+). According to figure 1A, the donor construct does not contain the IRFP reporter. Do the authors postulate that both px330 and donor vectors are co-transfected? Please clarify.

Reviewer is correct. Only Cas9/guide vector includes an IRFP expression cassette. Donor vector is not labeled. We co-transfected a ratio of donor : px330 vectors of 3:1 and subsequently selected for IRFP+ cells. We have amended this error in the revised version. Thanks for pointing it out.

In figure 2A, the PDO line-of-origin of the clone (named clone #1) used for the characterization is not clear. In addition, the characterization of LGR5+ cells should be detailed further for the second (#7) PDO (instead of only stating that the results were validated in a second PDO line). In particular, details should be given about the proportion of cells that are GFP+ and their gene expression profile.

We characterized two independent PDO7 clones (#1 and #2) bearing correct LGR5-GFP integrations. Data shown in Figure 2 correspond to PDO7 clone #1 whereas the characterization of PDO7 clone#2 is included in Supplementary Figure 2. The GFP+ tumor cell population was enriched in stem cell-like cells in both PDO7 derived clones and the distribution of LGR5-GFP+ and differentiated tumor cells in xenografts was also equivalent in both clones (Figure 2 and Figure EV3).

Extending this analysis to additional PDOs, representative of distinct classes of colorectal cancer, would greatly strengthen the manuscript.

We included the characterization of PDO6 in Figure EV1, which was derived from a different patient (the mutational profile of this sample is described in table 1). Our data shows that, similarly to PDO7, LGR5-GFP+ cell population from PDO6 was enriched in tumor cells that express the intestinal stem cell signature. The domains occupied by LGR5-GFP+ cells and by the differentiated compartment in xenografts were similarly distributed as in PDO7, i.e. it included MUC2+ intermingled within LGR5-GFP+ glands whereas the KRT20+ domain was complementary to that occupied by LGR5-GFP+ cells. We also show that LGR5-GFP+ purified from xenografts formed

organoids with highest efficiency than LGR5-GFP⁻ cells (Figure EV 1J). Unfortunately, PDO6 is poorly tumorigenic and only grew in mice upon inoculation of millions of cells, which precluded experiments of tumor initiation from isolated populations.

In figure 2H, I understand that organoids generated from LGR5-GFP⁻ cells contain a number of GFP⁺ cells similar to that of organoids from LGR5-GFP⁺ cells. This should be described in the text and commented upon in terms of implication for the nature of cancer stem cells and their biology. This comment also applies to figure 2I.

Following the reviewer suggestion, we quantified the number of LGR5-GFP⁺ cells present in xenografts derived from LGR5-GFP⁺ or LGR5-GFP⁻ cells and indeed found equivalent proportions and range of intensities (Figure 2J). In this revised version, we also provide evidence that LGR5⁻ cells generate xenografts that display similar phenotypic diversity than those produce by LGR5⁺ cells, including stem-like LGR5-GFP⁺ cells and differentiated LGR5-GFP⁻ tumor cells that express KRT20 or MUC2 (Figure 2K). Whereas these findings suggest that some LGR5-GFP⁻ cells are capable of regenerating the phenotypic diversity of the tumor of origin (and therefore that there is some degree of plasticity in the CRC hierarchy), it is not possible to rule out that the purification a small fraction of LGR5-GFP⁺ cells within the LGR5-GFP⁻ gate. A proper assessment of cell plasticity awaits upcoming experiments of lineage tracing from LGR5-GFP⁻ cell population. We have discussed these ideas in the revised version of the manuscript.

In figure 3D, why is one of the two organoids not recombined? Does this reflect a different genotype, low recombination efficiency or LGR5-independent organoid growth (this could be easily tested by double staining ER and TOM)?

The picture in Figure3D showed organoids that arose after passaging a culture that had been previously treated with 4-hydroxytamoxifen. We tried to exemplify the fact these secondary cultures contained organoids that were generated by either recombined cells (TOM⁺/BFP⁻) or non-recombined cells (TOM⁻/BFP⁺). We do agree with the reviewer that the image might be somewhat misleading. To avoid confusion, we have now replaced this image by another corresponding to an organoid 10 days after 4-hydroxytamoxifen treatment that has not been passaged and therefore it is formed by a mixture of recombined TOM⁺/BFP⁻ and non-recombined TOM⁻/BFP⁺ cells.

In LGR5-GFP knock-in PDOs every single organoid contained GFP⁺ cells, which argues against LGR5⁺ cell-independent growth. Furthermore, knock-in PDOs are monoclonal (single cell derived) and therefore all cells within the culture share identical genotypes including a CreERT2 cassette integrated at the LGR5 locus. Thus, the TOM mosaic organoids very likely reflect low recombination efficiency.

Given the number of LGR5-GFP⁺ cells and the number of TOM⁺ cells present at early time points, we have calculated a recombination efficiency of about 5-10% in both *in vitro* and *in vivo* experiments (see below). We suspect that this relatively low recombination efficiency may be due to suboptimal expression of the LF2A-creERT2 cassette (which is also commonly observed in knock-in mice) yet we cannot exclude epigenetic silencing of the targeted locus in some tumor cells. Nevertheless, the relatively low recombination frequency facilitated clonal analysis at single cell level in tumors, which is the gold standard to describe stem cell behavior (Blanpain and Simons, 2013). Our experience with anti-ER antibodies to assess creERT2 expression is that they only work in an overexpression setting. Accordingly, despite we tried to perform the experiment suggested by the reviewer it failed, probably due to low expression.

In the same paragraph, I don't understand the sentence "Analysis of xenografts 96h after induction with tamoxifen revealed the appearance of a TOM⁺ side population, which expressed LGR5 mRNA (Supplementary Fig 4) supporting tracing from the LGR5⁺ cell population". If TOM⁺ cells express the LGR5 mRNA, this supports the presence of LGR5⁺ cells in the xenograft rather than tracing from these cells (which should produce a majority of LGR5⁻, differentiated cells). Please correct or clarify.

Two combined effects explain that 96h after tamoxifen induction, the TOM⁺ cell population is still enriched in LGR5 mRNA:

- i. Our lineage tracing experiments indicate that the progeny of LGR5+ cells does not differentiate immediately but rather differentiation progresses over several weeks (Figure 3J). This finding is also supported by the observation that LGR5+ cells accumulate in clusters (Figure 2B) whereas KRT20+ cells occupy adjacent domains.
- ii. A proportion of LGR5+ cells remain quiescent (Figure 4) while expressing stem cell genes.

By the way, what proportion of the total cells is this category? Is it similar to that of LGR5-GFP+ cells?

Based on the frequency of LGR5-GFP-hi cells of about 4% (Figure 2D and Figure EV 3D) and the number of TOM+ cells arising 96h hours post-tamoxifen calculated by flow cytometry (Figure EV 4B), we roughly estimated 1 recombination event in every 10 to 20 LGR5-GFP-hi cells present in xenografts. In xenograft sections, recombined TOM+ cells after 96 hours represented about 0.5% of all epithelial cells, which also fits well with an estimation of about 10% of recombined LGR5-GFP+ cells. We obtained equivalent recombination efficiencies *in vitro*. We have included this information in the revised version.

In figure 3H, I agree that the results are compatible (i.e. do not contradict) with the hypothesis that LGR5 activity sustains the tumor growth. However, an experiment with a PDO line engineered to delete LGR5+ cells (for instance diphtheria toxin-based) would be much more convincing. In its absence, the sentence needs to be reworded.

We agree with this reviewer that cell ablation experiments could help confirm our hypothesis yet because of time constrains of this revision, we think they are beyond the scope of the present manuscript. Following his/her suggestion, we have rewritten this part to convey a more conservative message on the role of LGR5+ cells in human CRC.

In figure 3J, the dynamics of TOM+ cell differentiation is confusing. What are the cells produced from LGR5+ cells that do not display differentiation markers? Do they still express LGR5? This may result in much higher proportions of LGR5-GFP+ cells in the reporter experiments.

As we discuss above, TOM+ cells retain the expression of LGR5 and of other stem cell marker genes at least during the 96h that follow recombination implying that the progeny of LGR5+ cells do not undergo rapid differentiation. In this revised version we show that the LGR5-/Ki67+ cells express early markers of the absorptive lineage including FABP1 (Fatty acid binding protein 1) and SI (Sucrase Isomaltase) (Figure 4E). KRT20, which is expressed during terminal differentiation in the normal intestinal epithelium, accumulate in TOM+ clones only after several days of induction with tamoxifen and it is upregulated in LGR5-/Ki67- cells (Figure 4E). We thus hypothesize that LGR5-/Ki67- cells may represent proliferative progenitors undergoing progressive differentiation. This pattern is reminiscent of that present in the normal mucosa as previously proposed by several labs (Dalerba et al., 2011; Merlos-Suarez et al., 2011).

In figure 4C, it seems to me that two thirds of the LGR5-GFP cells express KI67-RFP. Why do the authors state that "a large proportion of LGR5-GFP+ population did not express KI67-tagRFP2"? If this is because the authors anticipated more KI67-expressing LGR5-GFP cells, based on their knowledge of the mouse stem cell population, this should be explained.

The reviewer is correct in his/her appreciation. In the normal intestinal mucosa, about 90% of all LGR5+ ISCs express KI67 and contribute progeny over time (Barker et al., 2007; Basak et al., 2014; Schepers et al., 2012). Therefore, the observation of higher frequency of LGR5+/KI67- cells (20-50% of all LGR5+ cells) in CRC is unexpected. We have now clarified this aspect in the discussion.

The kinetics of tumor cell turnover seems slower than in the healthy tissue. As this contrasts with some previous reports, this should be discussed in more detail. Alternatively, perhaps the authors could estimate the efficiency of Cre-mediated recombination within tumors?

As mentioned above we estimate a frequency of recombination *in vivo* of about 10%. It is important to consider that whereas we only traced a relatively small number of LGR5+ cells, the progeny of these LGR5+ cells scales proportional to tumor growth. This observation fits well with a model in

which the activity of multiple LGR5+ cells sustains tumor expansion yet we cannot rule out than some LGR5- cells display equivalent behavior and contribution to growth.

In the normal intestinal mucosa, most epithelial cells are replaced in a weekly basis. Cell loss is compensated by cell production in homeostatic conditions. The ratio of tumor cell turnover in CRC (generation of new cells minus cell loss over time) cannot be directly inferred from our data but our preliminary observations suggest that tumor growth results from the activity of LGR5+ cells combined with extended lifespans of progenitors and differentiated tumor cells compared to their normal counterparts (and possibly by increase of numbers of tumor stem cells undergoing self-renewing divisions according to expansion of the LGR5+ domains compared with the crypts). In the normal mucosa, ISCs divide virtually everyday (Barker et al., 2007; Snippert et al., 2010). Thus, it is unlikely that tumor cells divide much faster than normal ISCs. Winton and colleagues performed lineage tracing in adenomas (Kozar et al., 2013). They concluded that LGR5+ tumor cells in these tumors divide with a frequency similar to that of normal ISCs albeit few contribute over long term to sustain tumor growth. These and other aspects about the mode of tumor growth in human CRC are obviously very interesting but will require accurate quantitative clonal analysis and mathematical modeling, which is beyond the scope of the current manuscript.

In summary, although this manuscript needs a few improvements, the two major breakthroughs presented, i.e. (1) the technology for genome editing of human CRC cells and (2) the function of LGR5 within human CRC cells and the identification of a LGR5+;KI67- tumor cell population, make an important contribution in the field.

We thank the reviewer for the positive opinion and hope to have addressed his/her major concerns.

Referee #3 (Comments on Novelty/Model System):

The data here are interesting and novel and the statistics are fine. It is interpretation that is at issue currently.

Referee #3 (Remarks):

This is an interesting paper that describes development of methodology to test functionally "cancer stem cell" properties using cutting-edge techniques involving patient-derived organoids, CRISPR/Cas9 induced lineage tracing and cell cycle probes, and sequential xenografting. In all, it is an interesting set of methodologies, with some interesting findings. There are some key overinterpretations and lack of interpretations that should be rectified. There are a number of other smaller deficiencies. More or less in order of import:

1) The authors make comparisons of CRC stem cell hierarchy and kinetics to that of normal cells: "Clones generated by LGR5+ cells contained MUC2+ and KRT20+ cells (Fig 3I) yet we found that the frequency of differentiated cells in each clone increased over time (Fig 3J). Therefore, loss of the stem cell program and gain of differentiation traits in CRC is a relatively slow process compared to normal intestinal epithelium." How do these results compare with normal human colonic epithelium? The authors should cite and describe the kinetic stem/differentiated cell studies of this sort in normal epithelium that have been performed and can be used to compare to the current xenografted CRC studies.

We thank the reviewer for pointing this out. In the normal intestinal mucosa, the progeny of ISCs express differentiation markers early (about 48h) after leaving the crypt base while they migrate towards the surface. These differentiated cells die at the top of the crypts in about 5-7 days. Therefore, our data indicates that differentiation in CRC occurs with a slow kinetics compared to the normal colonic mucosa. We have clarified this concept in this revised version.

2) It is not clear why the authors think that some LGR5+ cells produced few progeny. First, what is this based on? The fact that there maintains some single and double clones? These could be artifacts of tissue cutting (they may be connected to bigger colonies above or below the plane of section) or it could be simply that the tamoxifen induction is slow and/or leaky with new clones popping up over time.

The reporter cassette is not leaky as shown by the absence of clones in untreated animals. We have made clear this point in the revised version. Furthermore, we observed single cells and small clones even 56d after tamoxifen in xenografts that have been transplanted to secondary untreated mice, thus ruling out that small clones resulted from remaining tamoxifen.

The reviewer is correct in his/her appreciation that small clones and individual cells could simply represent artifacts of our 2D analysis. To test this possibility, we have generated 3D reconstruction from multiple tissue sections at 28 days post-tamoxifen. This analysis confirmed the existence of labeled individual cells and small clones that are not connected to larger clones, which implies that marked *Lgr5* cells generated few progeny over this period. We have included these data in the Figure EV5 and the Movie EV1 of the revised version.

At any rate, it isn't clear that simply being Ki-67- implies any long-term state of quiescence. Or rather, perhaps the authors should define quiescence. There would seem to be several parameters involved: eg, how long G0/G1 and G2 phases are for the average LGR5+ cell. Perhaps that parameter is simply stochastic with broadly distributed normal curve? Another parameter is how long the Ki67-tag and mRNA signal perdure relative to phases of the cell cycle. One could easily envision a scenario, where all LGR5+ cells are part of essentially the same pool, each cell with normally (bell-shaped curve) distributed chance of entering the cell cycle, and there is a certain perdurance of Ki67 such that some of these cells at any time don't label with Ki67. In this scenario, there are no special, separate, quiescent LGR5+ cells, just a certain likelihood at any given time point that some of the population will be caught without detectable Ki67. If you could follow individual cells, perhaps many of these, 24h later would now be Ki-67 positive. The gene expression profiles do not really help with this analysis, as most depicted genes (except Ki67, of course) are relatively similar. *Aurkb* is decreased in Ki67-, but this is a G2/M transcript. *Cdkn1a*, on the other hand, is also a proliferation marker, and it is decreased in both Ki67+ and ki67-, but it is a G1-S transition gene. In short, it is not easy to conclude any kind of quiescent population in these *Lgr5*+/*Ki67*- cells without understanding the cell cycle kinetics of LGR5 and Ki67. If the authors would really like to make this case of a dedicated quiescent cell population, then studies of colony-forming potential would be useful. Or even simply taking the + and - populations from the organoids and seeing if they have any immediate differences or can interconvert.

To our understanding, there are two important aspects regarding the suitability of Ki67 as a marker of proliferation;

- 1) Ki67 is a nuclear protein which is expressed in all active parts of the cell cycle (G1, S, G2 and mitosis), but absent in resting cells (G0)(Scholzen and Gerdes, 2000). In contrast to many other cell cycle-associated proteins, the Ki67 antigen is consistently absent in quiescent cells and is not detectable during DNA repair processes. Thus, the presence of Ki67 antigen is strictly associated with the cell cycle (Scholzen and Gerdes, 2000). In fact, Ki67 expression is commonly used in pathology laboratories to assess the number of proliferative cells in tissues and tumors. In the intestinal crypts, 90% of all LGR5+ ISCs are labeled by Ki67 antibody(Basak et al., 2014). The Clevers lab generated knock-in mice expressing a Ki67-RFP fusion protein, which enabled the isolation of cycling (KI67-RFP+) and of non-cycling differentiated cells (Ki67-RFP-) in the intestinal epithelium (Basak et al., 2014). Of note, authors confirmed that vast majority (90%) of LGR5+ cells were Ki67-RFP+. We copied this strategy and engineered PDOs that express a fusion protein between Ki67 and RFP from the endogenous locus. Unlike in normal crypts, we found that a substantial proportion of LGR5+ cells were Ki67-RFP- (Figure 4C), an observation that is further backed by Ki67 immunohistochemistry on LGR5-GFP knock-in PDOs (Figure 4B).
- 2) Whereas lack of Ki67 expression is good marker of cells that are not cycling, expression of Ki67 may not always indicate that the cell is actively dividing. For instance, Winton and colleagues use DNA label retaining assay to identify a subpopulation of Ki67+ intestinal cells that are slow proliferating or quiescent (Buczacki et al., 2013).

To reinforce further our conclusions, we have now included the following data:

- We have analyzed cell cycle profiles of LGR5+/Ki67+ and LGR5+/Ki67- purified from xenografts (figure 4D). We confirmed that a large proportion of cells in G1/G0 phase of the cell cycle within the LGR5+/Ki67- population.
- We include additional regulators of the cell cycle, such as FOXM1 and UBE2C, all of which are downregulated in LGR5+/Ki67- population (Figure 4E). It is also important to note that there is an overall downregulation of the proliferation signature expressed by crypt progenitors (GSEAs in Figure 4G).
- We build signatures of genes up- and downregulated in normal LGR5+/Ki67+ and LGR5+/Ki67- crypt cells from transcriptomic profiles of LGR5-GFP/Ki67-RFP mice published by Basak et al (Basak et al., 2014). We found that these genesets were enriched in LGR5+/Ki67+ and LGR5+/Ki67- CRC cells respectively (Figure 4G), suggesting that tumor and normal LGR5+/Ki67- share some features.

As for the question of reversibility, we agree with the reviewer about the possibility that LGR5+/Ki67+ and LGR5+/Ki67- cells can interconvert and that therefore that cell cycle arrest could be transient in tumors. We have also commented on this issue in this revised version. Nevertheless, our clonal analysis indicates that a many LGR5+ cells contribute few progeny, which fits in well with the presence of LGR5+ cells in a state of inactivity over several weeks. Also, there is robust evidence in the literature about the existence of dormant stem cell in CRC from works that use lentiviral marking of individual tumor cell populations to study clonal expansion. In particular, Dick and colleagues showed that oxaliplatin treatment selectively favored the survival of dormant clones that became dominant after therapy (Kreso et al., 2013). Glimm and colleagues observed expansion of dormant cell clones upon CRC xenograft passaging (Dieter et al., 2011).

3) Similarly, the authors ignore plasticity with Lgr5- cells also being able to become Lgr5+. For example, this seems to be the basis of the claim that because cells that had been Lgr5+ (tomato+) at one point grows at the same rate as the overall population, this means that "tumor growth is largely the result of LGR5+ cell activity." One could certainly envision other interpretations to explain these results. For one, many of the cells in the tumor (LGR5+ and -) might divide symmetrically, and the LGR5+ cells might simply have a rate of division that is faster than the other cells'. To rule this out, strict bookkeeping of LGR5+ clones in terms of which cells are LGR5+ and which not would have to be done (currently not possible with no LGR5 antibodies). In short, why are we to assume KRT20+/LGR5- cells can't themselves divide? And they might also be plastic and become LGR5+ themselves, which may happen in Fig. 2H. I think it's better not to assume too much that differentiation goes only in an LGR5+ to negative direction. That isn't even sure in vivo in normal animals. Similarly, Fig. 2H seems to show that Lgr5-EGFP- cell-derived organoids eventually generate the same proportion of EGFP+ cells. Does that mean EGFP- cells can give rise to EGFP+ cells or that the few LGR5+ cells that were sorted into the GFP- fraction eventually proliferated sufficiently to "catch up" with the EGFP+ population? This is a key question because it would indicate the Lgr5- population is plastic as occurs when Lgr5+ cells are lost in normal tissue (eg following irradiation).

In sum: The paper largely reaches the conclusion that Lgr5+ lineage marks a stem cell population within a tumor. This cell lineage gives rise to a population of differentiated and differentiated-like cell populations within xenografts. If this is the true hierarchy, then why can EGFP-negative cells form organoids AND tumors? The characterization of tumors from such cells appear deficient. This is an interesting finding because one can speculate about other scenarios: plasticity where an EGFP-negative cell can occasionally (or facultatively) be recruited back to be a "stem cell," or alternative hierarchies where multiple "stem cell" lineages exist, or where Lgr5+ cells are an intermediate population, and another cell within the EGFP-negative population is the stem cell. The knock-in allele could also be sporadically silenced, and would need to be verified on single EGFP-negative cells by RT-PCR to determine if they are expressing Lgr5. Again, a plasticity scenario may be plausible in light of data presented in Figure 2H, where tumors derived from EGFP-negative cells were dissociated and plated to form organoids, which then contain EGFP-positive cells. Thus, EGFP-negative cells gave rise to EGFP-positive cells. Discounting this phenomenon (as the authors appear to do) is dangerous if it perpetuates a framework where one might think that therapeutic

elimination of Lgr5+ cells would kill the tumor - not true if plasticity between lineages occur. This isn't really surprising with transformed cells, which frequently change phenotypic states (EMT, MET, quiescence, etc.). A discussion of these data and these phenomenon must be included.

As indicated by this reviewer, our experiments do not rule plasticity between LGR5+ and LGR5- cells (and in fact, we do believe that plasticity occurs between these populations in established tumors). We have tried to convey this message in the discussion of this revised version.

In addition, we now provide data showing that LGR5- cells generate xenografts that display similar phenotypic diversity than those produced LGR5+ cells, including equivalent numbers of LGR5-GFP+ cells as well as LGR5-GFP- cells that express the differentiation markers KRT20 or MUC2. Whereas these observations strongly suggest that some LGR5-GFP- cells are capable of regenerating the phenotypic diversity of the tumor, we cannot exclude the possibility that a small fraction of LGR5-GFP+ cells was purified within the LGR5-GFP- gate. A proper assessment of plasticity and the contribution of differentiated cells to long-term tumor growth await upcoming experiments of lineage tracing from LGR5-GFP- cell population.

5) Also, perhaps the authors should discuss what it means that clone #1 had "few mutations compared to the parental population"? Did 64 de novo mutations occur in culture relative to the original patient isolate? Are the organoids changing that much, constantly? Can these modified PDOs be kept indefinitely? What about the increasing tumor burden implied by the large number of mutations vs. parental strain? These issues must be detailed in a paper whose impact will be determined by how well future users will be able to take advantage of these techniques.

Variant calling in exome sequencing data was performed with the widely used algorithm Mutect2. It has a false positive rate of about 6.4 mutations per Mb (Kroigard et al., 2016), which with the range of mutations that we found in our comparison. Prompted by the reviewer criticism, we reanalyzed the data using a different well-established algorithm – VarScan2-, which has a similar error rate (Kroigard et al., 2016). Comparison between Mutect2 and VarScan2 analyses showed that only 17 of the 64 missense/non-sense mutations between parental and knock-in PDOs were consistent. Of these 17, only 1 was predicted to be of high impact. In other words, a many of the mutations identified were likely false positive imposed by the methodology of analysis. We have included these data in the revised version (Appendix Table S3).

As for the stability of the organoids, in a recent study the lab of Toshiro Sato (Japan), performed exome sequencing in a small subset of CRC organoids after prolonged culture (Fujii et al., 2016). For the microsatellite stable CRC organoid analyzed (i.e. the same type of CRC used in our study), they found 9 non-synonymous mutations occurring after 6 months in culture, which is within the range that we encountered in edited PDOs. Other indirect measurements performed by the Clevers lab also suggest that the genomes of CRC organoids remain relatively stable over time (Weeber et al., 2015).

Smaller problems:

Why is there no discrete population of EGFP+ cells in so many FACS plots like fig. 1B,D; 2E,H? The histological assessment in Fig. 1D and 2A-D seems much more as expected with clear positive and negative cells. What do controls that weren't transfected at all look like? In other words, is this a question of background fluorescence or something else?

Yes, PDOs are autofluorescent as shown in the wide spectrum of emission of parental unmodified organoids both in vitro (Figure 1D) and in vivo (Figure 2D). LGR5-GFP-hi cells were sufficiently bright to enable their isolation by FACS. However, autofluorescence imposed limitations in our ability to purify GFP-low cells, which were largely captured in the GFP-negative fraction. In tissue sections, we used anti-GFP antibodies to label LGR5-GFP+ cells. These stainings illustrate better the heterogeneous expression of the LGR5-GFP reporter.

Similarly, the increase in Egfp in qPCR is relatively modest (Fig. 1E). Is this because the EGFP- cells actually have EGFP transcripts that aren't turned over? What Ct levels were these negative cells say vs. non-transfected cells? The lack of decreased protein in "negative" cells might be due to GFP perdurance, which might lead to cells that are negative for mRNA be sorted in the EGFP+ category.

That would lead to less differential expression also but of a different sort, where the numerator (EGFP+) cells are actually much lower in EGFP mRNA expression than would be expected and drives down fold increase. What Ct levels were there in EGFP^{Hi} vs. negative vs. non-transfected?

The LGR5-GFP reporter strategy yielded better results in *in vivo* experiments than in cultured organoids. Differential expression was in the range of 20 fold for both LGR5 and EGFP in cells purified from xenografts (Figure 2E, Figures EV11 and 3E), which I would not describe as modest. However, as noticed by this reviewer, *in vitro* differences in expression were less pronounced (in the range of 3-4 fold, Figure 1E and Figure EV 1D). We found that LGR5-negative cells purified from xenografts express equivalent levels of GFP mRNA yet they have about 4-8 fold less LGR5 and SMOC2 mRNA than their *in vitro* counterparts. Therefore; i. LGR5-negative cells in organoids retain some expression of ISC genes suggesting that lack of terminal differentiation *in vitro*. ii. Rather than persistence of GFP protein in LGR5-GFP negative cells, our observation suggests that GFP mRNA is more stable than the mRNA of endogenous ISC genes.

As for the specific question of the reviewer regarding GFP expression in negative versus non-transfected PDOs, Ct values are 24 versus 36 in cells isolated in cells from organoid cultures. As mentioned above, expression of GFP in the negative fraction is explained by the inclusion of LGR5-low cell population as a result of background autofluorescence (please read above).

Similarly, there seems to be a great deal of fluorescence in control populations in Fig. 1I, though, of course, the red channel is obviously substantially elevated in Ki67-TagRFP.

Thanks for pointing this out. We realized that the FACS plots shown in Fig 1I correspond to tumor cells after click-it reaction used to detect EdU incorporation. Fluorescence in control organoids corresponds to background of the reaction. We have now included the plots corresponding to the same population before staining, which shows virtually no background fluorescence.

P4, lines 19-20: IRFP+ marks the px330 vector (not the donor)

Reviewer is correct. Only Cas9/guide vector includes an IRFP expression cassette. Donor vector is not labeled. We co-transfected donor : px330 vectors in 3:1 ratio and subsequently selected for IRFP+ cells. We have corrected this error in the revised version. Thanks for pointing this issue out.

Why do the authors say that some glands are Lgr5+ and KRT20+, which implies that the Lgr5 and KRT glands are separate entities? Wouldn't the authors expect that glands would be mixes of "stem cell"-like cells and the KRT20+ and MUC2+ cells that arise from them? If not, how would KRT20+/LGR5- glands come about?

We do observe that LGR5-GFP+ areas contain few differentiated cells (most of which express MUC2+) whereas the majority of KRT20+ cells reside in complementary (LGR5-) domains rather than intermingled with LGR5+ cells. In some sections, we captured the boundary between the two domains, i.e. a tumor gland with LGR5-/KRT20+ area immediately adjacent to LGR5+/KRT20-cells (examples in Figure 2K) but in many others these two marker genes appear to label distinct tumor glands. This distribution was also observed in knock-in PDO6 (Figure EV1F) and might be common to several but not every CRC as we previously suggested (Merlos-Suarez et al., 2011). We propose that this pattern recreates the organization of the normal mucosa as previously proposed by several labs ((Dalerba et al., 2011; Merlos-Suarez et al., 2011)), characterized by progenitor and differentiated cells residing in distinct compartments. Moreover, this finding fits in well with the notion that LGR5+ tumor cells generate progeny that do not differentiate immediately and may suggest that distinct tumor niches facilitates stem or differentiation states. We have tried to convey these ideas in this revised version of the manuscript.

What is the control in Fig. 4C? It corresponds to the parental population and therefore it is not CRISPR targeted.

REFERENCES

Barker, N., van Es, J.H., Kuipers, J., Kujala, P., van den Born, M., Cozijnsen, M., Haegebarth, A., Korving, J., Begthel, H., Peters, P.J., *et al.* (2007). Identification of stem cells in small intestine and colon by marker gene Lgr5. *Nature* 449, 1003-1010U1001.

- Basak, O., van de Born, M., Korving, J., Beumer, J., van der Elst, S., van Es, J.H., and Clevers, H. (2014). Mapping early fate determination in Lgr5⁺ crypt stem cells using a novel Ki67-RFP allele. *EMBO J* *33*, 2057-2068.
- Blanpain, C., and Simons, B.D. (2013). Unravelling stem cell dynamics by lineage tracing. *Nat Rev Mol Cell Biol* *14*, 489-502.
- Buczacki, S.J., Zecchini, H.I., Nicholson, A.M., Russell, R., Vermeulen, L., Kemp, R., and Winton, D.J. (2013). Intestinal label-retaining cells are secretory precursors expressing Lgr5. *Nature* *495*, 65-69.
- Dalerba, P., Kalisky, T., Sahoo, D., Rajendran, P.S., Rothenberg, M.E., Leyrat, A.A., Sim, S., Okamoto, J., Johnston, D.M., Qian, D., *et al.* (2011). Single-cell dissection of transcriptional heterogeneity in human colon tumors. *Nature biotechnology* *29*, 1120-1127.
- Dieter, S.M., Ball, C.R., Hoffmann, C.M., Nowrouzi, A., Herbst, F., Zavidij, O., Abel, U., Arens, A., Weichert, W., Brand, K., *et al.* (2011). Distinct types of tumor-initiating cells form human colon cancer tumors and metastases. *Cell Stem Cell* *9*, 357-365.
- Fujii, M., Shimokawa, M., Date, S., Takano, A., Matano, M., Nanki, K., Ohta, Y., Toshimitsu, K., Nakazato, Y., Kawasaki, K., *et al.* (2016). A Colorectal Tumor Organoid Library Demonstrates Progressive Loss of Niche Factor Requirements during Tumorigenesis. *Cell Stem Cell* *18*, 827-838.
- Kozar, S., Morrissey, E., Nicholson, A.M., van der Heijden, M., Zecchini, H.I., Kemp, R., Tavare, S., Vermeulen, L., and Winton, D.J. (2013). Continuous clonal labeling reveals small numbers of functional stem cells in intestinal crypts and adenomas. *Cell Stem Cell* *13*, 626-633.
- Kreso, A., O'Brien, C.A., van Galen, P., Gan, O.I., Notta, F., Brown, A.M., Ng, K., Ma, J., Wienholds, E., Dunant, C., *et al.* (2013). Variable clonal repopulation dynamics influence chemotherapy response in colorectal cancer. *Science* *339*, 543-548.
- Kroigard, A.B., Thomassen, M., Laenkholm, A.V., Kruse, T.A., and Larsen, M.J. (2016). Evaluation of Nine Somatic Variant Callers for Detection of Somatic Mutations in Exome and Targeted Deep Sequencing Data. *PLoS One* *11*, e0151664.
- Merlos-Suarez, A., Barriga, F.M., Jung, P., Iglesias, M., Cespedes, M.V., Rossell, D., Sevillano, M., Hernando-Momblona, X., da Silva-Diz, V., Munoz, P., *et al.* (2011). The intestinal stem cell signature identifies colorectal cancer stem cells and predicts disease relapse. *Cell Stem Cell* *8*, 511-524.
- Schepers, A.G., Snippert, H.J., Stange, D.E., van den Born, M., van Es, J.H., van de Wetering, M., and Clevers, H. (2012). Lineage tracing reveals Lgr5⁺ stem cell activity in mouse intestinal adenomas. *Science* *337*, 730-735.
- Scholzen, T., and Gerdes, J. (2000). The Ki-67 protein: from the known and the unknown. *J Cell Physiol* *182*, 311-322.
- Snippert, H.J., van der Flier, L.G., Sato, T., van Es, J.H., van den Born, M., Kroon-Veenboer, C., Barker, N., Klein, A.M., van Rheenen, J., Simons, B.D., *et al.* (2010). Intestinal crypt homeostasis results from neutral competition between symmetrically dividing Lgr5 stem cells. *Cell* *143*, 134-144.
- Weeber, F., van de Wetering, M., Hoogstraat, M., Dijkstra, K.K., Krijgsman, O., Kuilman, T., Gadellaa-van Hooijdonk, C.G., van der Velden, D.L., Peeper, D.S., Cuppen, E.P., *et al.* (2015). Preserved genetic diversity in organoids cultured from biopsies of human colorectal cancer metastases. *Proc Natl Acad Sci U S A* *112*, 13308-13311.

Thank you for the submission of your revised manuscript to EMBO Molecular Medicine. We have now received the enclosed report from the reviewer who was asked to re-assess it. As you will see s/he is now globally supportive and I am pleased to inform you that we will be able to accept your manuscript pending the amendments previously requested of you by separate cover and the additional following ones:

- 1) As per our Author Guidelines, the description of all reported data that includes statistical testing must state the name of the statistical test used to generate error bars and P values, the number (n) of independent experiments underlying each data point (not replicate measures of one sample), and the actual P value for each test (not merely 'significant' or 'P < 0.05'). If you prefer, you can list the P values in a separate appendix table. In such case please make sure you amend your manuscript to include the appropriate callouts.
- 2) Please make sure that accession numbers for microarray and human xenograft data are clearly featured in the manuscript text.
- 3) The manuscript must include a statement in the Materials and Methods identifying the institutional and/or licensing committee approving the experiments, including any relevant details (like how many animals were used, of which gender, at what age, which strains, if genetically modified, on which background, housing details, etc). We encourage authors to follow the ARRIVE guidelines for reporting studies involving animals. Please see the EQUATOR website for details: <http://www.equator-network.org/reporting-guidelines/improving-bioscience-research-reporting-the-arrive-guidelines-for-reporting-animal-research/>. Please make sure that ALL the above details are reported. I note that some information is in the manuscript and some in the checklist. Please make sure ALL the information is also in the manuscript.
- 4) We encourage the publication of source data, with the aim of making primary data more accessible and transparent to the reader. Would you be willing to provide a PDF file per figure that contains the original, uncropped and unprocessed scans of all or at least the key gels used in the manuscript and/or source data sets for relevant graphs? The files should be labeled with the appropriate figure/panel number, and in the case of gels, should have molecular weight markers; further annotation may be useful but is not essential. The files will be published online with the article as supplementary "Source Data" files. If you have any questions regarding this just contact me.

I look forward to seeing a revised form of your manuscript as soon as possible. The sooner we receive it, the sooner we can proceed with acceptance for publication.

***** Reviewer's comments *****

Referee #1 (Comments on Novelty/Model System):

As stated before

Referee #1 (Remarks):

The authors emphasized in their response that the primary aim of the manuscript is the description of a new methodology that allows cell fate mapping of primary human cancer cells in vivo. As I pointed out in my first assessment I think it should be published because of its novelty and relevance for the field. Additional data were added suggesting bidirectional plasticity of LGR5 expression very much in line with very recent publications from the Sato and de Sauvage groups in Nature. Given the compelling evidence now from three independent studies I strongly believe that reviewer concerns were convincingly addressed, further experimental work is not needed and publication should not be delayed.

2nd Revision - authors' response

07 April 2017

Authors made requested editorial changes.

Corresponding Author Name: Eduard Batlle

Journal Submitted to: The EMBO Journal

Manuscript Number: